# EgoTactile: Learning Grasp Pressure for Everyday Objects from Egocentric Video

**Yuan Zeng** [1]  **Yujia Shi** [2 3]  **Tiao Tan** [1]  **Xingting Li** [1]  **Yaqi Qin** [4]  **Zongqing Lu** [1]  **Wenming Yang** [1]  **Jing-Hao Xue** [5]  **Qingmin Liao** [1]

## Abstract

Estimating full-hand grasp pressure from egocentric video is critical for immersive VR and robotic manipulation, yet dense tactile sensing often relies on intrusive hardware. Existing vision-based methods predominantly rely on planar surfaces or fingertip contacts, failing to generalize to complex 3D object interactions. Therefore, we introduce EgoTactile, a benchmark pairing egocentric video with full-hand pressure supervision for diverse everyday objects, incorporating a bare-hand transfer subset to enable generalization to natural scenarios. Leveraging this benchmark, we first establish EgoPressureFormer as a discriminative baseline. Beyond this, to explicitly address the uncertainty in partial observations, we propose EgoPressureDiff, a conditional diffusion framework that adapts a large-scale pre-trained video diffusion backbone. By combining rich world knowledge priors with a Physically-Informed Feature Rectification layer to inject semantic constraints, our approach effectively infers plausible contact patterns and resolves visual-physical ambiguities. Extensive experiments demonstrate that our method achieves superior performance on the benchmark and robust transferability to in-the-wild scenarios. Our project page is available at https://egotactile.github.io.

[1]Shenzhen International Graduate School, Tsinghua University, Shenzhen, China [2]School of Computer Science and Technology, Harbin Institute of Technology, Shenzhen, China [3]Department of Network, Pengcheng Laboratory, Shenzhen, China [4]JQ Industries, Qingdao, China [5]Department of Statistical Science, University College London, London, United Kingdom. Correspondence to: Qingmin Liao <liaoqm@tsinghua.edu.cn>.

*Proceedings of the 43rd International Conference on Machine Learning*, Seoul, South Korea. PMLR 306, 2026. Copyright 2026 by the author(s).

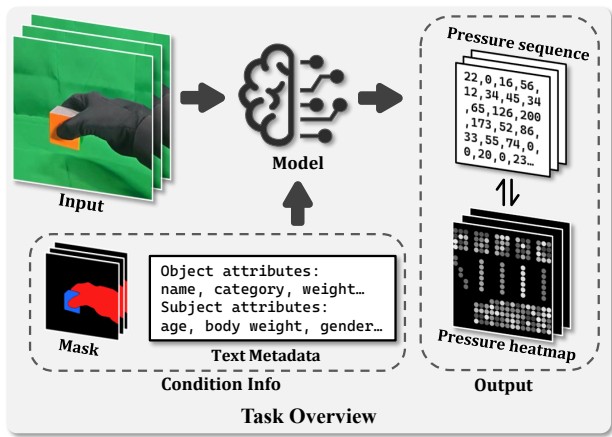

*Figure 1.* Task overview. Given an RGB clip of a human-object interaction, the model predicts contact pressure, optionally incorporating auxiliary condition information to reduce physical ambiguities. The output is represented as either a sparse sensor sequence or a dense spatial heatmap, which are convertible.

## 1. Introduction

Dense pressure sensing is critical for enabling immersive virtual reality (VR) interactions and robust robotic manipulation, yet accessing such high-fidelity tactile signals remains expensive to scale and intrusive (Yuan et al., 2017; Lambeta et al., 2020). Consequently, learning pressure from vision has emerged as a promising non-intrusive alternative. Pioneering works validated this approach by exploiting visual cues such as soft-tissue deformation and cast shadows. To capture these fine-grained appearance changes, PressureVision (Grady et al., 2022) employs a CNN-based encoder-decoder to regress a pixel-aligned pressure map directly from RGB pixels. Extending this, EgoPressure (Zhao et al., 2025) utilizes a transformer to estimate pressure as a UV texture directly on the hand mesh.

However, despite these advancements, existing approaches face two fundamental limitations. First, they predominantly rely on planar surfaces where contact regions are continuously visible, enabling straightforward pixel-to-pressure mapping but failing to generalize to complex 3D geometries. Second, they often simplify interactions to fingertip-only

contacts (Grady et al., 2024), neglecting the dynamic multi-finger and palm pressure distributions inherent in full-hand grasping. Motivated by these gaps, we propose a novel task: Egocentric 3D Grasp Pressure Prediction. Unlike prior works constrained by planar visibility or simplified contact patterns, this task explicitly targets the learning of dynamic hand pressure on diverse 3D objects. We ground this task in the egocentric video setting due to its centrality in VR, wearable applications, and robot imitation learning (Grauman et al., 2022; Damen et al., 2018; Streli et al., 2024; Banerjee et al., 2025). While this viewpoint captures nuanced hand-object interactions, it inherently introduces severe perceptual and physical complexities that render simple regression methods inadequate. As shown in Figure 1, this task serves as a new frontier for modeling human force strategies and learning transferable tactile representations.

Yet, establishing this frontier is non-trivial and requires overcoming three challenges: (i) The Data Gap. Synchronized full-hand pressure data for 3D objects under controlled settings remains scarce. While OpenTouch (Song et al., 2025) offers 3D object interaction data, its in-the-wild nature introduces environmental noise that hinders rigorous variable analysis. (ii) Occlusion and Incomplete Observation. The assumption of pixel-to-pressure alignment breaks down in egocentric grasping since critical contact regions are often hidden. Under such partial observability, deterministic regression becomes ill-posed and tends to yield conservative, blurred predictions. (iii) Physical Ambiguity. Visually identical objects may possess vastly different physical attributes such as weight or fill state. As noted in tactile studies (Sundaram et al., 2019; Seo et al., 2024), resolving these ambiguities requires integrating non-visual physical priors beyond raw RGB cues.

To tackle these challenges, we present a comprehensive solution comprising a benchmark and two complementary baselines. First, to close the Data Gap, we establish *Ego-Tactile*, a benchmark providing synchronized full-hand pressure supervision for 63 everyday objects. Furthermore, we introduce a novel "bare-hand" transfer subset, which coordinates synchronized gloved and bare-hand grasps to create a weakly-paired supervision paradigm, enabling generalization to natural, sensor-free scenarios. Second, to tackle Occlusion and Incomplete Observation, we construct two complementary baselines, *EgoPressureFormer* as a discriminative reference and, crucially, our primary contribution *EgoPressureDiff*, a conditional diffusion framework. We argue that generative modeling is essential for partial observability, as unlike deterministic regression, a diffusion model can represent the multimodal nature of the solution space under uncertainty. By adapting a pre-trained Stable Video Diffusion (SVD) (Blattmann et al., 2023) backbone, our model leverages a "world model" prior to infer plausible contact patterns even when visual cues are occluded. Finally, to

disentangle Physical Ambiguity from visual appearance, we propose a *Physically-Informed Feature Rectification (PIFR)* layer. Integrated within the diffusion backbone, this module explicitly injects structured semantic constraints (e.g., object weight, stiffness) to calibrate force magnitudes. This ensures that the generated pressure maps are not only visually coherent but also physically faithful to attributes that are invisible to the camera.

Our main contributions: (i) We propose egocentric 3D grasp pressure prediction task and release EgoTactile, a benchmark pairing egocentric video with full-hand pressure for 3D objects, including a bare-hand subset for broader applications. (ii) We propose EgoPressureDiff, a conditional diffusion framework that leverages generative priors to resolve occlusion and uncertainty. (iii) Through comprehensive experiments and visualizations, we demonstrate our method's superior performance on the benchmark and its robust generalization to unconstrained, real-world scenarios.

## 2. Related Work

### 2.1. Visual-Tactile Datasets

Understanding the fine-grained physical dynamics of hand-object interaction is a core challenge in computer vision and robotics. Regarding visual pressure estimation datasets, PressureVision (Grady et al., 2022) and EgoPressure (Zhao et al., 2025) infer pressure from RGB but are limited to planar surfaces. While PressureVision++ (Grady et al., 2024) introduces natural objects, it only supervises fingertips, lacking the full-hand contact distribution required for analyzing complex grasps. We summarize existing visual pressure estimation datasets in Table 1. In the domain of egocentric full-hand touch and activity datasets, OpenTouch (Song et al., 2025) aligns in-the-wild video with tactile maps, but its uncontrolled setting hinders precise variable analysis, while ActionSense (DelPreto et al., 2022) captures multi-modal kitchen activities but lacks dense pressure supervision for specific object grasping. Furthermore, visual-tactile pre-training datasets like $M^2VTP$ (Liu et al., 2024), VTDexManip (Liu et al., 2025), and humanoid datasets (Kwon et al., 2025) typically rely on sparse or binarized tactile signals for representation learning or teleoperation, rather than high-fidelity, continuous pressure estimation from human vision. Finally, purely tactile studies on full-hand tactile signatures (Sundaram et al., 2019; Ceprián-Bernal & Pérez-González, 2021; Seo et al., 2024) demonstrate that pressure signatures encode object identity and weight, yet they lack the synchronized egocentric visual context essential for training vision-based prediction models. In contrast, EgoTactile bridges these gaps by providing large-scale, paired egocentric video and full-hand pressure data in controlled scenarios, enabling rigorous analysis of pressure-visual correlations under occlusion and varying physical attributes.

*Table 1.* Comparison with representative visual pressure estimation datasets. Unlike prior works limited to planar surfaces or fingertip-only supervision, EgoTactile provides synchronized full-hand pressure on 3D objects, accompanied by Subject attributes (age, body weight, body fat rate, gender) and Object attributes (material types, weight). The complete dataset comparison table is provided in Appendix D.

| Dataset | Viewpoint | Interaction Surface | Pressure Coverage | Scale | Participants | Objects | Subj. Attr. | Obj. Attr. |
|---|---|---|---|---|---|---|---|---|
| PressureVisionDB (Grady et al., 2022) | Exo | Planar surface | – | 3.0M frames | 36 | – | ✗ | ✗ |
| ContactLabelDB (Grady et al., 2024) | Exo | Diverse surfaces | – | 2.9M frames | 51 | 106 | ✗ | ✗ |
| EgoPressure (Zhao et al., 2025) | Ego+Exo | Planar surface | – | 4.3M frames | 21 | – | ✓ | ✗ |
| **EgoTactile (Ours)** | **Ego** | **3D objects** | **Full hand (162 taxels)** | **319k frames** | **12** | **63** | ✓ | ✓ |

## 2.2. Visual Hand Pressure Estimation

Recent approaches infer contact pressure from RGB images by leveraging visual cues such as soft-tissue deformation and cast shadows. PressureVisionNet (Grady et al., 2022) pioneered this direction using a CNN-based encoder-decoder to estimate pixel-wise pressure from a single image. To enable generalization to uninstrumented "in-the-wild" surfaces, PressureVision++ (Grady et al., 2024) introduced a semi-supervised framework leveraging weak supervision from prompted "contact labels". It employs adversarial domain adaptation to align feature distributions between fully labeled sensor data and weakly labeled diverse samples. Furthermore, EgoPressure (Zhao et al., 2025) proposed PressureFormer, a transformer-based architecture that estimates pressure as a UV texture map on the hand mesh, thereby enforcing 3D geometric consistency through differentiable rendering. More recently, EgoPressDiff (Zeng et al., 2026) extends this line of work by formulating egocentric hand pressure estimation as conditional video diffusion, generating UV-pressure maps with multimodal geometric guidance from RGB, depth, pose, and MANO vertices. Nevertheless, it remains focused on the EgoPressure planar-touch setting, whereas EgoTactile addresses full-hand pressure estimation for occluded 3D object grasps with additional physical ambiguities. While we explore a non-spatially aligned video transformer baseline, EgoPressureFormer, to mitigate occlusion, we find that deterministic regression remains insufficient for resolving such ambiguities (e.g., object weight). Consequently, we propose EgoPressureDiff, a generative diffusion framework that leverages world priors and physical information to infer pressure under uncertainty.

## 3. Task and Pressure Formulation

### 3.1. Task Definition

**Inputs&outputs.** As shown in Figure 1, we address egocentric grasp pressure prediction by mapping a monocular RGB video clip, optionally augmented with auxiliary conditions, to a dynamic pressure sequence or heatmap. Formally, given an egocentric video stream, we construct a fixed-length clip $\mathbf{X} = \{\mathbf{x}_t\}_{t=1}^T$ by uniform temporal sampling, where $\mathbf{x}_t \in \mathbb{R}^{H \times W \times 3}$ denotes the $t$-th RGB frame. To supply structured priors beyond raw pixels, we may optionally provide conditioning signals $\mathbf{C}$, which include: (1)

per-frame segmentation masks $\mathbf{m}_t$; and (2) clip-level text attributes summarizing object properties (e.g., weight, surface material) and subject attributes (e.g., gender, age, body fat).

The goal is to predict the pressure distribution aligned to each frame $t$. We consider two inter-convertible representations, the ground-truth glove pressure sequence $\mathbf{p}_t \in \mathbb{R}^M$ (with $M = 162$) and a corresponding hand pressure heatmap $\mathbf{H}_t \in \mathbb{R}^{S \times S}$ defined on a canonical canvas of size $S$. Accordingly, a model outputs either $\widehat{\mathbf{p}}_t \in \mathbb{R}^M$ or $\widehat{\mathbf{H}}_t \in \mathbb{R}^{S \times S}$. The heatmap is single-channel with non-hand regions set to 0, and the two forms are deterministically converted for unified evaluation.

### 3.2. Pressure Representation

**Normalization.** To handle the high dynamic range and sensor noise, we normalize raw pressure readings to $[0, 1]$. Specifically, values below a noise threshold $p_{\min} = 5\text{N}$ (Newton) are zeroed out, and the result is scaled by a robust upper bound $p_{\max} = 200\text{N}$ (99.9% quantile), with outliers clipped.

**Bidirectional Mapping.** We utilize two convertible representations: the sparse pressure sequence $\mathbf{p}_t \in \mathbb{R}^M$ and the dense hand heatmap $\mathbf{H}_t \in \mathbb{R}^{S \times S}$. To bridge these modalities, we precompute a fixed linear rendering operator $A$ based on the canonical hand geometry, which projects sequences into heatmaps via Gaussian diffusion. Conversely, to recover discrete sensor values from predicted heatmaps for evaluation, we solve the inverse problem via Ridge regression. The detailed mathematical formulation of the operator $A$ and the inverse solver is provided in Appendix A.2.5.

### 3.3. Evaluation Metrics

To make evaluation comparable across models that predict either the sequence $\widehat{\mathbf{p}}_t \in \mathbb{R}^M$ or the heatmap $\widehat{\mathbf{H}}_t \in \mathbb{R}^{S \times S}$, we convert all predictions into the per-frame pressure sequence form. Concretely, for methods generating heatmaps, we recover $\widehat{\mathbf{p}}_t$ via the deterministic inverse conversion detailed in Appendix A.2.5, while methods that directly output $\widehat{\mathbf{p}}_t$ are utilized as-is. All metrics are computed on the aligned sequences $\{(\mathbf{p}_t, \widehat{\mathbf{p}}_t)\}_{t=1}^T$.

Following PressureVision (Grady et al., 2022), we report four standard metrics on the pressure sequences: (i) Tempo-

ral accuracy of contact/non-contact over time, (ii) Contact IoU measuring overlap of thresholded contact patterns, (iii) Volumetric IoU measuring magnitude-aware overlap (via min/max aggregation), and (iv) MAE measuring absolute pressure error. While the above metrics effectively capture temporal consistency and magnitude accuracy, they do not explicitly evaluate the geometric precision of pressure distribution within specific anatomical regions. To address this, we additionally report the Part-wise Center-of-Pressure (CoP) Error, which quantifies the spatial deviation of predicted pressure centroids on fingertips and the palm. The detailed mathematical formulation of this metric is provided in Appendix C.2.

## 4. EgoTactile Benchmark

### 4.1. Data Collection Protocol

**Sensors and Setup.** Our acquisition platform synchronizes egocentric RGB videos with full-hand pressure signals (Figure 2a). Visual data is captured by a DJI Action 5 Pro ($1280 \times 720$, 30 fps), employing both neck and head mounts to emulate realistic wearable viewpoints (Figure 2c-d). Tactile signals are recorded via a glove with $M{=}162$ sensors (0–350 N range, sampled at $\sim$17 Hz). Data collection takes place in a controlled green-screen environment to avoid visual clutter. Crucially, to ensure robustness, we randomize lighting conditions and object poses across sessions.

**Two Acquisition Modes.** Our data collection is designed to support both effective learning and real-world transfer. First, learning an accurate vision-to-pressure mapping benefits from reliable, temporally aligned supervision that can be collected at scale. Second, to support generalization to real-world egocentric interactions, it is desirable for the model to learn from bare-hand visual inputs. We therefore adopt two complementary acquisition modes: the Gloved-hand set and the Bare-hand set.

*Gloved-hand set.* In this setting, participants wear the tactile glove on the grasping hand, and the gloved hand is visible in the egocentric view during interaction. This setting provides paired samples of egocentric video and synchronized pressure signals, enabling direct supervised learning.

*Bare-hand set.* To evaluate and improve generalization to real-world egocentric videos where gloves are not worn, we additionally construct a bare-hand set. In this setting, as shown in Figure 2b, the hand visible in the egocentric video is bare, while another hand wearing the tactile glove performs a synchronized grasp of the same object outside the camera's field of view to provide pressure labels. To improve motion synchrony between the visible bare-hand action and the gloved-hand action, participants follow a metronome-guided rhythm

that standardizes the timing of the five-stage sequence (approach/contact/grasp/release/retreat). To avoid models exploiting a single fixed tempo as a shortcut, the metronome frequency is varied across objects for the same participant. The resulting data provide a weakly paired supervision signal: bare-hand video with synchronized pressure labels.

*Bare-hand protocol validation.* Because the bare-hand set is weakly paired rather than perfectly aligned, we further validate its consistency with a dual-glove study that simulates the same protocol. Across 3 subjects, 5 held-out objects, and 10 repetitions per object, the two pressure streams show small temporal discrepancies, with average contact-onset and release gaps of 105 ms and 118 ms, respectively, corresponding to less than two frames on our synchronized 15 Hz timeline. The two hands also achieve 82.7% contact IoU with small CoP and force differences. These results indicate that the weak pairing is imperfect but sufficiently consistent for transfer evaluation. Detailed results are provided in Appendix E.1.

**Text Metadata.** For each recorded clip, we store structured metadata that can be used as optional conditioning signals and for controlled analyses. Object-side attributes include: object name, object category, weight, surface material, and load state (filled/empty) when applicable (e.g., bottles with varying fill levels). Subject-side attributes include: age, body weight, body fat rate, gender, hand length and dominant hand. All subject attributes are self-reported and optional. We store metadata in anonymized form and do not release any personally identifiable information.

### 4.2. Dataset Diversity and Statistics

**Object and Subject Diversity.** As detailed in Table 2, our benchmark covers 63 everyday objects across 7 high-level categories. This semantic diversity translates into a wide spectrum of physical interactions: as illustrated in Figure 2f, the recorded peak forces vary significantly across object instances, covering both delicate and power grasps. To capture the impact of human factors, we recruit 12 participants with balanced gender distribution and diverse physiological attributes (e.g., body fat, hand length). The influence of these individual differences is evident in Figure 2h, which highlights distinct average pressure profiles across participants. Furthermore, the aggregated contact probability and average pressure heatmaps (Figure 2e, g) confirm that our dataset captures dense, full-hand contact patterns, rather than sparse touches.

**Dataset Splits.** To benchmark both object generalization and cross-subject generalization, we define two evaluation protocols on the gloved-hand set. (1) Object-held-out split: we partition object instances into disjoint train/test sets, such that test objects are never seen during training. Unless

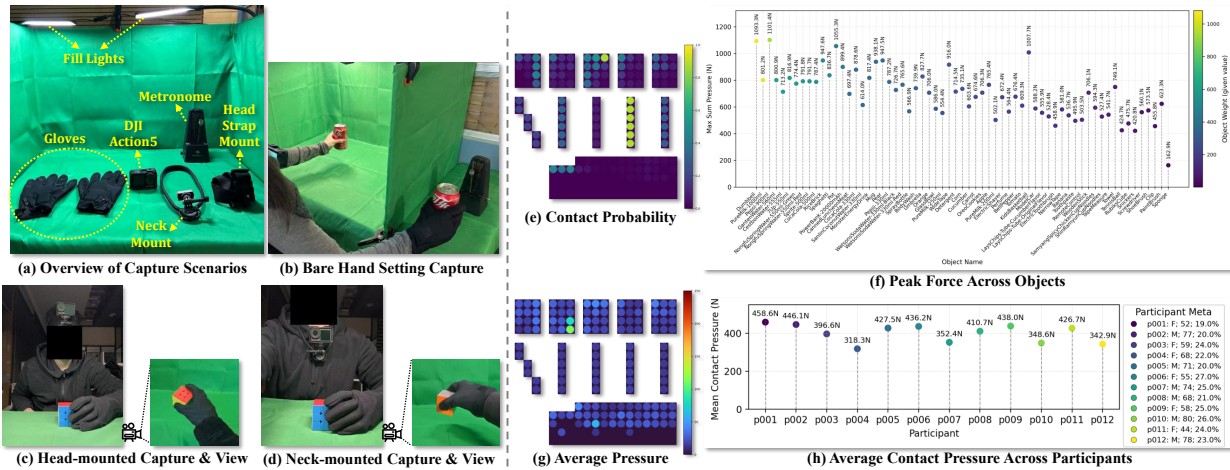

**(a) Overview of Capture Scenarios** **(b) Bare Hand Setting Capture** **(e) Contact Probability** **(f) Peak Force Across Objects**

**(c) Head-mounted Capture & View** **(d) Neck-mounted Capture & View** **(g) Average Pressure** **(h) Average Contact Pressure Across Participants**

*Figure 2.* Data collection setup and dataset statistics. Left: Our capture environment features controlled lighting and a green-screen background (a), and (b) illustrates the data collection scenario for bare-hand setting. To ensure viewpoint diversity and realistic transfer, we capture data using both head-mounted (c) and neck-mounted (d) cameras. Right: Statistics of the collected data, including hand contact probability (e), average pressure heatmaps (g) and force magnitude distributions across objects (f) and participants (h).

*Table 2.* Dataset Diversity Statistics. Left: Distribution of the 63 objects across 7 categories. Right: Detailed demographics of the 12 participants, explicitly covering physical attributes (weight, body fat) that influence grasp stability.

| Object Distribution | | Subject Demographics | |
|---|---|---|---|
| **Category** | **# Count** | **Attribute** | **Statistics** |
| Packaging | 26 | **Total Subjects** | **12** |
| Daily Household | 11 | **Gender** | 6 Male / 6 Female |
| Fruits&Veg | 9 | **Age** | $26.2 \pm 2.1$ (years) |
| Tools&Electronics | 6 | **Body Weight** | $65.3 \pm 11.5$ (kg) |
| Sports&Toys | 5 | **Body Fat Rate** | $23.3 \pm 2.6$ (%) |
| Kitchenware | 3 | **Hand Length** | $18.2 \pm 0.7$ (cm) |
| Office Supplies | 3 | **Dominant Hand** | 12 Right / 0 Left |
| **Total Objects** | **63** | | |

*Table 3.* Dataset statistics and evaluation protocols. We report the total scale for both acquisition modes and the specific train/test splits used for benchmarking geometric generalization (Object-Held-Out) and subject robustness (Subject-Held-Out). The total number of unique participants is 12, as there is an overlap of 2 subjects between the two settings.

| Set | Protocol / Split | # Obj. | # Subj. | # Clips | Dur. (h) |
|---|---|---|---|---|---|
| | ***Total Collected*** | **63** | **12** | **768** | **5.82** |
| **Gloved** | *Protocol 1: Object-Held-Out* | | | | |
| | Train | 58 | 11 | 638 | 4.81 |
| | Test | 5 | 11 | 55 | 0.42 |
| | *Protocol 2: Subject-Held-Out* | | | | |
| | Train | 63 | 9 | 630 | 4.74 |
| | Test | 63 | 2 | 63 | 0.46 |
| **Bare** | *Protocol: Object-Held-Out* | | | | |
| | Train | 20 | 3 | 60 | 0.49 |
| | Test | 5 | 3 | 15 | 0.12 |

otherwise specified, we hold out a fixed portion of object instances for testing. (2) Subject-held-out split: we partition participants into disjoint train/test groups, such that test participants are never seen during training. This protocol evaluates robustness to individual differences in grasping strategy and force scale. For the bare-hand set, we do not introduce a subject-held-out protocol, as its primary purpose is cross-appearance evaluation rather than cross-subject robustness. Table 3 summarizes the final sample counts for each split.

## 5. Baselines

Most prior visual pressure estimation methods model pressure as a pixel-aligned prediction problem, implicitly assuming that the pressure-bearing regions are visible and can be spatially aligned to image pixels (Grady et al., 2022; 2024; Zhao et al., 2025). This assumption is frequently violated in our benchmark, as self-occlusion and object occlusion often

hide the contact regions, making explicit pixel-to-pressure alignment ill-posed. In addition, grasp pressure is inherently temporal with structured phases such as approach, contact onset, stable holding, and release, and single-frame prediction cannot fully capture the dynamics of contact transitions and force modulation. Motivated by these gaps, we construct two baselines that (i) explicitly model temporal dynamics and (ii) avoid relying on dense pixel-level alignment. We first describe a discriminative temporal encoder-decoder baseline and then introduce a generative diffusion-based baseline in the next subsection.

### 5.1. Baseline I: EgoPressureFormer

As the discriminative representative, we introduce EgoPressureFormer, a sequence predictor built upon the TimeSformer backbone (Bertasius et al., 2021). Directly addressing the limitation of pixel-aligned methods under occlusion,

this model abandons dense heatmap regression. Instead, it employs a query-based decoding mechanism where learnable sensor embeddings attend to spatiotemporal visual tokens to directly predict the pressure sequence for all $M$ sensors. This design effectively bypasses the need for explicit spatial alignment, allowing the model to infer pressure from global context even when contact regions are hidden. To mitigate the sparsity of contact events, the model is trained with a multi-task objective combining per-sensor classification and a frame-level contact gate. Further details regarding the architecture and training objective are provided in Appendix B.2.

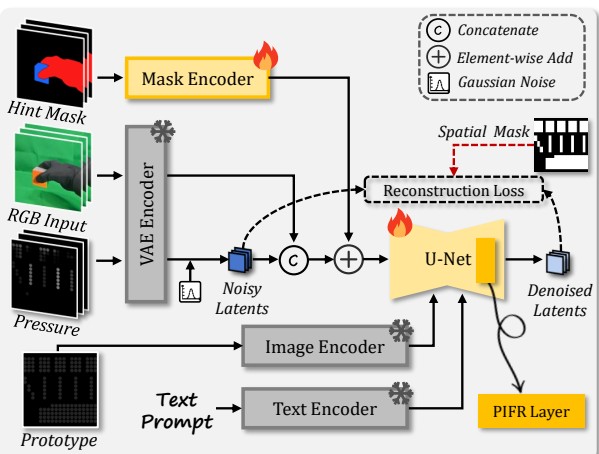

**EgoPressureDiff Training Pipeline**

*Figure 3.* We formulate pressure estimation as a diffusion process conditioned on egocentric RGB video. To resolve physical ambiguities, we incorporate multimodal guidance via: (i) a hint mask processed by a Mask Encoder, and (ii) text prompts and a prototype heatmap injected through the proposed PIFR Layer.

### 5.2. Baseline II: EgoPressureDiff

**Overview.** In egocentric grasps, pressure estimation is often ill-posed: when most contact regions are occluded by the object, multiple pressure distributions may be consistent with the same visible pixels. A strong generative prior is therefore desirable to complete plausible contact patterns under missing observations. We thus propose EgoPressureDiff, a generative baseline that adapts SVD (Blattmann et al., 2023) to pressure heatmap synthesis, leveraging its rich pre-trained interaction and world priors. As illustrated in Figure 3, EgoPressureDiff formulates pressure estimation as conditional diffusion and leverages multimodal conditioning to guide denoising under ambiguity. The core condition is the egocentric RGB video, which provides dynamic cues about contact transitions and hand–object interaction. To align modalities and stabilize training, we encode both the RGB clip and the pressure heatmap sequence into the same latent space using the VAE (Van Den Oord et al., 2017) that comes with SVD, and feed the noisy pressure latent together with the RGB latent via channel concatenation to the de-

noising model. In addition, we inject three complementary sources of guidance: (i) hint masks as a structured spatial prior indicating likely interaction regions, (ii) text prompts encoding high-level physical and subject information to disambiguate force scale when appearance is insufficient, and (iii) a pressure-heatmap prototype serving as an anatomical topology prior, constructed by assigning a uniform pressure (100N) across the hand region to define plausible contact areas. Text and prototype conditions are encoded by CLIP text/image encoders (Radford et al., 2021), respectively, and injected through the PIFR layer, which will be detailed in the following paragraphs. For mask conditioning, we pass the mask images through a lightweight Mask Encoder, implemented as a shallow stack of convolutional blocks with SiLU activations (Elfwing et al., 2018), and add the resulting mask features to the latent input before denoising, thereby providing spatial guidance throughout the diffusion process.

**Physically-Informed Feature Rectification (PIFR) Layer.** While RGB videos and hint masks provide dynamic spatial cues, they fail to resolve force magnitude under physical ambiguity (e.g., identical objects with different weights). To address this, text prompts encode the necessary physical constraints but lack spatial precision. Complementing both, the prototype heatmap offers a stable anatomical topology prior, though it remains agnostic to pressure magnitude. As a result, naive fusion of these features often leads to physically implausible intensities or unstable conditioning, where the denoiser over-relies on the prototype and under-utilizes the text. Considering that SVD (Blattmann et al., 2023) is natively designed to accept image embeddings as conditions, we opt to use text to rectify the prototype representation rather than directly introducing it into the denoising process. We therefore introduce a simple rectification layer that calibrates the spatial prototype using text-conditioned physical attributes before conditioning SVD.

Figure 4b illustrates the mechanism of the PIFR layer. Let $\mathbf{z}$ denote an intermediate latent feature map in the SVD denoiser. We obtain two conditioning feature maps via cross-attention: a prototype-conditioned feature $\mathbf{z}^{proto}$ and a text-conditioned feature $\mathbf{z}^{text}$. Concretely, we use $\mathbf{z}$ as the query and attend to (a) prototype tokens derived from CLIP image embedding and (b) text tokens derived from CLIP text embedding. We then predict modulation parameters from $\mathbf{z}^{text}$:

$$[\boldsymbol{\gamma}, \boldsymbol{\beta}] = \mathcal{F}_{mod}(\mathbf{z}^{text}), \tag{1}$$

where $\mathcal{F}_{mod}(\cdot)$ is a lightweight mapping network producing a scale factor $\boldsymbol{\gamma}$ and a shift factor $\boldsymbol{\beta}$. To stabilize training, $\mathcal{F}_{mod}$ is zero-initialized so that the module starts as an identity transform. We rectify $\mathbf{z}^{proto}$ by an affine transformation:

$$\mathbf{z}^{cond} = \mathbf{z}^{proto} \odot (1 + \boldsymbol{\gamma}) \oplus \boldsymbol{\beta}, \tag{2}$$

where $\odot$ and $\oplus$ denote element-wise multiplication and

*Table 4.* Quantitative comparison on the Gloved-hand benchmark. We report results under two protocols: Object-Held-Out and Subject-Held-Out. Metrics include Temporal Accuracy (Temp Acc.), Contact IoU (C-IoU), and Volumetric IoU (V-IoU), all reported in percentage (%), while MAE is measured in Newtons (N). CoP Error is reported in grid units (Euclidean distance on the sensor layout coordinate grid). Random Guesser is a marginal-probability sanity baseline evaluated on Object-Held-Out. Ablation variants (bottom rows) are evaluated on the Object-Held-Out split to analyze component contributions.

| | Object-Held-Out | | | | | Subject-Held-Out | | | | | Params | FPS |
|---|---|---|---|---|---|---|---|---|---|---|---|---|
| **Method** | **Temp Acc.↑** | **C-IoU↑** | **V-IoU↑** | **MAE↓** | **CoP↓** | **Temp Acc.↑** | **C-IoU↑** | **V-IoU↑** | **MAE↓** | **CoP↓** | **(M)** | **(fps)** |
| Random Guesser | 41.8 | 7.6 | 4.1 | 15.4 | 30.9 | – | – | – | – | – | – | – |
| PressureVision(Grady et al., 2022) | 65.2 | 24.5 | 16.8 | 9.2 | 12.5 | 61.5 | 21.2 | 14.3 | 10.5 | 14.2 | 589 | 28.2 |
| EgoPressureFormer | 84.5 | 36.8 | 26.5 | 6.2 | 7.5 | 80.2 | 32.5 | 22.8 | 7.5 | 8.8 | 3182 | 6.5 |
| *w/ regress raw pressure* | 68.5 | 22.4 | 14.2 | 12.8 | 15.6 | – | – | – | – | – | 3182 | 6.5 |
| *w/o frame-level contact head* | 72.8 | 29.5 | 21.6 | 7.8 | 9.2 | – | – | – | – | – | 3095 | 7.1 |
| **EgoPressureDiff** | **96.4** | **56.3** | **38.9** | **3.4** | **3.1** | **92.5** | **51.2** | **34.0** | **5.8** | **4.7** | 5978 | 2.8 |
| *w/o mask conditioning* | 94.2 | 51.5 | 35.8 | 3.9 | 4.2 | – | – | – | – | – | 5965 | 2.9 |
| *w/o prototype* | 95.1 | 47.8 | 33.2 | 4.2 | 5.5 | – | – | – | – | – | 5620 | 3.2 |
| *w/o text* | 95.8 | 54.5 | 30.5 | 5.1 | 3.3 | – | – | – | – | – | 5850 | 3.0 |
| *w/o PIFR (naive fusion)* | 95.5 | 53.8 | 31.9 | 4.8 | 3.5 | – | – | – | – | – | 5951 | 2.9 |
| *w/o spatial mask loss* | 95.0 | 53.2 | 36.5 | 3.8 | 3.9 | – | – | – | – | – | 5978 | 2.8 |

element-wise addition, respectively, and $\mathbf{z}^{cond}$ is the calibrated conditioning feature passed to the subsequent denoising blocks. Intuitively, $\mathbf{z}^{proto}$ preserves anatomically valid pressure topology (where pressure can appear), while $(\boldsymbol{\gamma}, \boldsymbol{\beta})$ inject magnitude constraints implied by physical attributes (how strong the pressure should be).

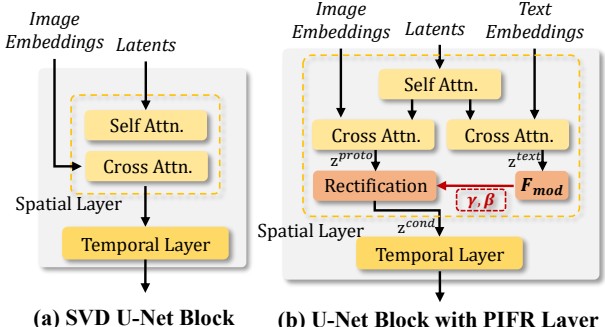

**(a) SVD U-Net Block**          **(b) U-Net Block with PIFR Layer**

*Figure 4.* (a) The original U-Net block of SVD. (b) Our proposed PIFR layer integrated into the U-Net block. Here, $\gamma$ and $\beta$ denote the scale and shift factors, respectively.

**Training objective.** As illustrated in Figure 3, we train EgoPressureDiff in latent space using the standard diffusion noise-prediction objective. To encourage the model to focus on the physically valid hand area, we incorporate a binary hand-region mask as a spatial prior in the loss computation. The training loss is formulated as a masked mean squared error:

$$\mathcal{L}_{\text{diff}} = \mathbb{E}\left[\left\|\mathbf{W} \odot \left(\boldsymbol{\epsilon} - \widehat{\boldsymbol{\epsilon}}\right)\right\|_2^2\right], \qquad (3)$$

where $\boldsymbol{\epsilon} \sim \mathcal{N}(\mathbf{0}, \mathbf{I})$ is the Gaussian noise added in the forward diffusion process, $\widehat{\boldsymbol{\epsilon}}$ is the noise predicted by the denoiser, and $\mathbf{W}$ is a binary spatial mask selecting the valid hand region on the canonical template.

## 6. Experiments

### 6.1. Experimental Setup

We evaluate baselines on Gloved-hand (Object/Subject-Held-Out), Bare-hand (Object-Held-Out), and realistic-scene settings, with all outputs unified to pressure sequences for evaluation. We additionally include a marginal-probability Random Guesser to verify that models do not merely exploit dataset-level contact statistics. Results for PressureFormer (Zhao et al., 2025) are not reported because its code is not publicly available at the time of submission. PressureVision++ (Grady et al., 2024) is not directly included because its core design relies on weak supervision. After removing the weak-supervision components, it reduces to a PressureVision-like supervised baseline. More implementation details are provided in Appendix C.1.

### 6.2. Gloved-hand Setting Performance

Table 4 summarizes the quantitative results on the Gloved-hand benchmark. We first observe that the marginal-probability Random Guesser performs substantially worse than all learned models. This confirms that full-hand grasp pressure cannot be recovered from dataset-level contact statistics alone, and that the learned models must exploit non-trivial visual-physical correlations from interaction videos. EgoPressureDiff demonstrates superior performance across all metrics. Notably, on the Object-Held-Out split, our method achieves a substantial margin compared to the strong baseline EgoPressureFormer, improving Volumetric IoU by +12.4% (26.5% → 38.9%). Furthermore, the CoP Error is reduced by more than half (7.5 → 3.1), indicating significantly higher precision in localizing pressure centroids. We attribute this improvement to the diffusion model's ability to infer plausible contact patterns even under severe self-occlusion, whereas deterministic regressors tend to output conservative, blurred predictions when visual

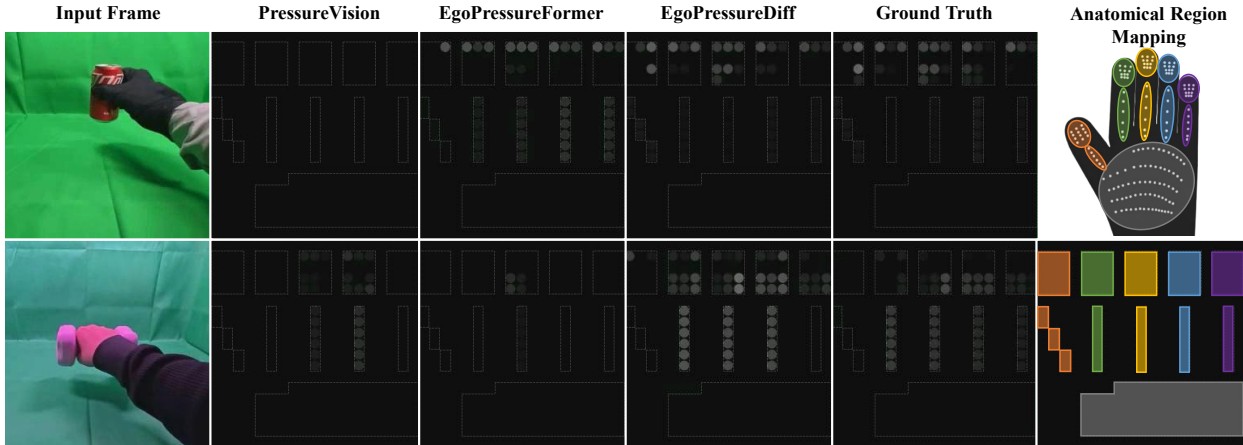

*Figure 5.* Qualitative comparison on Gloved-hand (Top row) and Bare-hand (Bottom row) settings. EgoPressureDiff generates spatially coherent pressure heatmaps with sharp contact peaks, accurately recovering individual fingertips even under self-occlusion (Row 1). In the Bare-hand transfer setting (Row 2), the fine-tuned EgoPressureDiff adapts to the appearance shift significantly better than baselines, which suffer from severe artifacts or under-estimation. For clarity, dashed hand outlines are overlaid on the heatmaps to indicate the hand contour. As shown in the Anatomical Region Mapping (right hand), the upper regions correspond to the fingers (from Thumb to Little finger, left to right), while the bottom region represents the Palm.

evidence is ambiguous, as qualitatively shown in Figure 5. Under the Subject-Held-Out protocol, all methods experience performance drops due to the domain shift caused by diverse hand shapes and grasping habits. While EgoPressureFormer suffers a noticeable degradation, our method maintains high performance, achieving 92.5% Temporal Accuracy and 51.2% Contact IoU. This resilience suggests that the learned structural priors and the injection of subject-specific text prompts effectively help the model generalize across individual physiological differences. For more qualitative visualization results, please refer to Appendix E.6.

**Ablation Study on EgoPressureFormer.** Replacing the bin classification with direct regression of raw sensor values (*w/ regress raw pressure*) leads to a drastic drop in C-IoU ($-14.4\%$) and nearly doubles the MAE ($6.2 \to 12.8$). This confirms that direct regression is ill-conditioned due to the high granularity and class imbalance of tactile signals. Similarly, removing the binary contact head (*w/o frame-level contact head*) degrades C-IoU to 29.5%. Since the dataset is dominated by non-contact frames, the explicit contact gate is essential to decouple the contact decision from the fine-grained pressure estimation.

**Ablation Study on EgoPressureDiff.** Removing mask conditioning (*w/o mask conditioning*) generally degrades all metrics. More critically, ablating the hand pressure prototype (*w/o prototype*) causes the CoP Error to spike from 3.1 to 5.5, the highest among all ablation variants. This explicitly confirms that the prototype serves as a vital anatomical topology prior. Removing the text prompt (*w/o text*) or removing the PIFR module (*w/o PIFR*) primarily impairs magnitude-sensitive metrics. Specifically, removing text

prompts drops V-IoU by 8.4% and increases MAE to 5.1 N, while spatial metrics like CoP remain relatively stable. While quantitative ablations verify the contribution of the PIFR layer, we further investigate its interpretability via counterfactual prompting experiments. We show that the model can decouple physical attributes from visual appearance by manipulating text prompts. A finer-grained text ablation further shows that object attributes are the dominant source of physical conditioning, while subject attributes provide complementary individual priors, especially under Subject-Held-Out evaluation. Detailed results and analysis are provided in Appendix E.4 and Appendix E.2.

**Inference efficiency.** EgoPressureDiff is slower than discriminative baselines due to iterative denoising. To assess deployability, we additionally perform latent consistency distillation (Song et al., 2023), using the original 25-step EgoPressureDiff as the teacher while preserving the same VAE, CLIP encoders, mask branch, prototype conditioning, and PIFR design. The 8-step student improves speed from 2.8 FPS to 6.9 FPS while retaining competitive performance (54.9% C-IoU vs. 56.3% for the teacher), and the 4-step student further reaches 10.3 FPS with a larger but still moderate drop. These results suggest that the current diffusion formulation can be substantially accelerated without redesigning the task formulation. Detailed speed-accuracy and resolution trade-offs are provided in Appendix E.3.

### 6.3. Bare-hand Transfer

Predicting pressure on bare hands is vital but lacks ground truth due to sensor occlusion. To address this, we adopt a Pre-training to Adaptation paradigm where models pre-trained on the gloved dataset are fine-tuned on the bare-hand

*Table 5.* Bare-hand transfer results.

| Method | Temp Acc.↑ | C-IoU↑ | V-IoU↑ | MAE↓ | CoP↓ |
|---|---|---|---|---|---|
| *Setting A: Zero-shot (Gloved → Bare)* | | | | | |
| PressureVision | 58.0 | 15.5 | 10.2 | 11.5 | 16.0 |
| EgoPressureFormer | 70.2 | 24.8 | 18.5 | 8.8 | 10.5 |
| **EgoPressureDiff** | **91.5** | **47.2** | **32.5** | **4.5** | **4.2** |
| *Setting B: Fine-tuned (Gloved → Bare)* | | | | | |
| PressureVision | 62.5 | 22.8 | 15.0 | 9.5 | 12.8 |
| EgoPressureFormer | 83.0 | 36.2 | 25.8 | 6.4 | 7.8 |
| **EgoPressureDiff** | **97.0** | **57.8** | **40.2** | **3.2** | **2.9** |

set. Direct transfer suffers from severe gap. As shown in Table 5 Setting A, discriminative baselines degrade drastically. In contrast, EgoPressureDiff exhibits remarkable robustness, maintaining a C-IoU of 47.2%, surpassing even the fine-tuned results of the discriminative baselines. This confirms that our diffusion backbone learns generalizable physical priors beyond surface appearance. Upon fine-tuning (Setting B), EgoPressureDiff achieves 57.8% C-IoU and 40.2% V-IoU, notably surpassing its own performance on the gloved setting shown in Table 4. We attribute this gain to the increased diversity and volume of data, which further unlocks the model's potential and suggests that the current data scale has not yet reached the learning capacity of the diffusion-based architecture. These results validate the practical viability of EgoPressureDiff for unconstrained scenarios. See Appendix E.6 for more visual results.

### 6.4. Robustness in Realistic Scenes

To quantify robustness beyond the controlled green-screen setting, we collect an additional realistic gloved-hand test set with pressure labels. It contains 30 egocentric clips across 10 everyday scenes, including kitchen, bathroom sink area, office desk, dining room, living room, and bedroom, with cluttered backgrounds and unconstrained lighting. All grasped objects are unseen during training and cover common household items such as a mug, water bottle, bowl, shampoo bottle, remote control, and stapler.

*Table 6.* Quantitative results in realistic gloved-hand scenes under the Object-Held-Out protocol. "Controlled" denotes the controlled Object-Held-Out benchmark in Table 4, while "In-The-Wild" denotes the newly collected realistic-scene test set. Δ denotes the change from Controlled to In-The-Wild.

| Method | Setting | C-IoU↑ | V-IoU↑ | MAE↓ | CoP↓ |
|---|---|---|---|---|---|
| PressureVision | Controlled | 24.5 | 16.8 | 9.2 | 12.5 |
| | In-The-Wild | 13.4 | 8.3 | 14.0 | 16.0 |
| | Δ (Change) | -11.1 | -8.5 | +4.8 | +3.5 |
| EgoPressureFormer | Controlled | 36.8 | 26.5 | 6.2 | 7.5 |
| | In-The-Wild | 23.7 | 16.1 | 8.9 | 10.7 |
| | Δ (Change) | -13.1 | -10.4 | +2.7 | +3.2 |
| **EgoPressureDiff** | Controlled | 56.3 | 38.9 | 3.4 | 3.1 |
| | In-The-Wild | 49.6 | 33.7 | 4.6 | 5.4 |
| | Δ (Change) | **-6.7** | **-5.2** | **+1.2** | **+2.3** |

As shown in Table 6, all methods degrade under realistic scene changes. However, EgoPressureDiff exhibits the smallest degradation and remains substantially stronger than the discriminative baselines. Compared with the controlled Object-Held-Out benchmark, its C-IoU and V-IoU decrease by only 6.7% and 5.2%, respectively, whereas PressureVision drops by 11.1% and 8.5%. EgoPressureDiff also shows the smallest increases in MAE and CoP error. These results suggest that the generative prior from the pre-trained video diffusion backbone improves robustness to background and illumination shifts, helping the model infer physically plausible contact patterns even in realistic scenes. Qualitative in-the-wild examples without ground-truth pressure labels are provided in Appendix E.5.

## 7. Limitations and Future Work

EgoTactile is designed to provide reliable full-hand pressure supervision, and therefore its main supervised split is collected in a controlled environment. Although our realistic-scene evaluation shows encouraging robustness, broader real-world coverage remains an important future direction. The bare-hand subset also provides weakly paired supervision rather than exact force labels for the visible hand. Our dual-glove validation indicates consistent timing and contact patterns, but residual mismatch may still affect fine-grained pressure estimation. In addition, stress tests on rare interaction patterns such as palm support, hook lift, and toss-and-catch reveal that current RGB-conditioned models remain limited when detailed hand pose and object 6D pose are not explicitly modeled. Finally, EgoPressureDiff is still slower than discriminative baselines, although latent consistency distillation substantially improves throughput. Future work will explore explicit geometric conditioning and adaptation to robot hands for scalable imitation learning.

## 8. Conclusion

In this work, we introduce EgoTactile, a benchmark bridging egocentric vision and full-hand tactile sensing for 3D objects, incorporating a bare-hand subset for transfer to natural scenarios. While we establish EgoPressureFormer as a discriminative baseline, we propose EgoPressureDiff to better address the ill-posed nature of pressure estimation under occlusion. This conditional diffusion framework leverages generative priors to resolve multimodal uncertainty. Crucially, the Physically-Informed Feature Rectification (PIFR) layer injects semantic constraints to ensure physically faithful predictions under visual ambiguity. Extensive experiments demonstrate our approach's superior performance and robust generalization in the wild. We believe this work provides a rigorous foundation for learning dense tactile representations, advancing capabilities in immersive VR and robotic manipulation.

## Acknowledgements

We thank the reviewers for the thoughtful discussion and feedback. This work was supported by the National Natural Science Foundation of China (U23B2030, Nos. 62311530100 and 62171251) and the Special Foundations for the Development of Strategic Emerging Industries of Shenzhen (No. KJZD20231023094700001).

## Impact Statement

This paper presents work whose goal is to advance the field of Machine Learning. There are many potential societal consequences of our work, none of which we feel must be specifically highlighted here.

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

# A. Dataset and Data Processing

## A.1. Benchmark Details

### A.1.1. OBJECT INVENTORY AND ATTRIBUTES

Our dataset includes a diverse collection of 63 everyday objects spanning 7 categories: Packaging, Daily Household, Fruits&Vegetables, Tools&Electronics, Sports&Toys, Kitchenware, and Office Supplies. These objects were selected to cover a wide range of weights (2g to 1082g), surface materials (plastic, metal, organic, ceramic, etc.), and stiffness properties. Table 7 lists the detailed attributes for all objects.

### A.1.2. PARTICIPANT DEMOGRAPHICS & PRIVACY

Data collection involved 12 participants (6 females, 6 males) aged between 23 and 29. All subjects are right-hand dominant. Table 8 summarizes their demographic profiles and hand dimensions.

**Privacy and Ethics.** This study was conducted with strict adherence to ethical guidelines. All participants provided written informed consent prior to data collection, explicitly allowing the use of their hand data for research purposes. To ensure privacy, we implemented the following protocols: (1) All data is strictly anonymized, with participants referred to only by IDs; (2) The egocentric camera angle is strictly focused on the hand-object interaction area, ensuring that no faces or identifiable biometric features are captured; (3) All recordings were conducted in a neutral laboratory environment to prevent the leakage of personal location information.

### A.1.3. GRASPING PROTOCOL

Our goal is to capture transferable human grasping patterns under everyday, habitual strategies, rather than scripted grasp taxonomies. Participants are instructed to grasp each object using their most familiar natural grasp, without being asked to switch grasp styles or apply prescribed force profiles. Each participant is required to use their dominant hand for grasping. For each object, each participant performs 5 repetitions. Each repetition contains a full approach $\rightarrow$ contact $\rightarrow$ grasp/hold $\rightarrow$ release $\rightarrow$ retreat cycle, taking approximately 6 seconds, resulting in roughly 30 seconds of interaction per object per participant. To mitigate fatigue effects, participants take a 5-minute break after every 15 minutes of recording. Trials are self-paced, with participants deciding when to start and end a trial via verbal cues (start/end utterances) to provide a consistent operational protocol across sessions.

### A.1.4. SYNCHRONIZATION PROTOCOL

Video and tactile streams have different native sampling rates (video: 30Hz; glove: 17Hz). We adopt a software synchronization strategy. Concretely, a master thread runs at a fixed rate of 15Hz, independent of the camera and glove acquisition threads. At each master tick, we (i) clear the video buffer and retain only the most recent frame, discarding older frames, and (ii) read the most recent glove measurement from a cached buffer. This yields aligned samples on a unified 15 Hz time axis via downsampling (video) and sample-and-hold (glove), which are then stored as synchronized pairs. Unless otherwise stated, all released benchmark clips and evaluations operate on this 15 Hz synchronized timeline.

### A.1.5. TRAIN/TEST SPLITS

To rigorously evaluate the generalization capability of our model, we define two evaluation protocols for both the Gloved-hand and Bare-hand subsets. For gloved-hand set, we evaluate performance under two protocols: (i) Object-Held-Out Protocol: We select 5 representative objects with varying shapes as the unseen test set: Apple, CocaCola-330ml, Corn, Dumbbell, and TennisBall. All sequences involving these objects are excluded from training. (ii) Subject-Held-Out Protocol: We select participant p007 and p011 as the unseen subjects. For bare-hand set, we employ an Object-Held-Out Protocol using the same set of unseen objects.

## A.2. Pressure Signal Processing and Representation

### A.2.1. FINGER-SEGMENT STABILIZATION

To improve temporal stability, we apply the glove firmware's deterministic sensor-aggregation protocol to the raw pressure readout. Specifically, we treat the frame-level pressure sequence $\mathbf{p}_t \in \mathbb{R}^M$ as the per-sensor measurement over $M$ sensing elements. The protocol produces a stabilized sequence $\widetilde{\mathbf{p}}_t \in \mathbb{R}^M$ by aggregating sensors on finger middle/proximal segments. For each finger $k \in \{1, \ldots, 5\}$, a six-sensor group $\mathcal{G}_k$ is predefined, and stabilized readings are obtained by replacing values within each group by their within-group mean:

$$\widetilde{\mathbf{p}}_t[i] = \begin{cases} \frac{1}{6} \sum_{j \in \mathcal{G}_k} \mathbf{p}_t[j], & \text{if } i \in \mathcal{G}_k \text{ for some } k, \\ \mathbf{p}_t[i], & \text{otherwise.} \end{cases} \quad (4)$$

This operation enforces a shared value across all indices in $\mathcal{G}_k$, suppressing segment-level noise while preserving the original indexing and dimensionality. Unless otherwise noted, we treat the stabilized signal $\widetilde{\mathbf{p}}_t$ as our ground truth and omit the tilde for simplicity in the rest of the paper.

*Table 7.* Complete object inventory for the **EgoTactile** dataset. Abbreviations: *Wt.* = Weight, *Mat.* = Surface Material, *plas* = plastic, *alum* = aluminum, *met* = metal.

| Items 1 – 31 | | | | | Items 32 – 63 | | | | |
|---|---|---|---|---|---|---|---|---|---|
| **Object** | **Category** | **Wt.(g)** | **Mat.** | **Fill** | **Object** | **Category** | **Wt.(g)** | **Mat.** | **Fill** |
| BodyWash | Daily Household | 314 | plastic | full | CocaCola-500ml | Packaging | 522 | plastic | full |
| Detergent | Daily Household | 251 | plastic | full | GantenWater-560ml | Packaging | 596 | plastic | full |
| ElectricShaver | Daily Household | 200 | plas/met | n/a | LaysChips-Tube-Cucumber | Packaging | 152 | paper | n/a |
| ElectricToothbrush | Daily Household | 143 | plas/rub | n/a | LaysChips-Tube-Original | Packaging | 151 | paper | n/a |
| Perfume | Daily Household | 192 | glass | full | MonsterEnergyDrink | Packaging | 352 | alum | full |
| ShoeBrush | Daily Household | 41 | wood/plas | n/a | NongfuSpring-550ml-G | Packaging | 573 | plastic | full |
| Sponge | Daily Household | 2 | foam | n/a | NongfuSpring-550ml-R | Packaging | 573 | plastic | full |
| SprayBottle | Daily Household | 337 | plastic | full | OreoCookies | Packaging | 221 | plastic | n/a |
| Toothpaste | Daily Household | 129 | plastic | full | Pepsi-330ml | Packaging | 347 | alum | full |
| Towel | Daily Household | 77 | fabric | n/a | Pepsi-900ml | Packaging | 944 | plastic | full |
| Umbrella | Daily Household | 292 | fab/met | n/a | PureMilk-1000ml | Packaging | 1065 | paper | full |
| PowerBank-20k | Electronics | 429 | met/plas | n/a | PureMilk-200ml | Packaging | 216 | paper | full |
| RemoteControl | Electronics | 116 | plastic | n/a | PureMilk-250ml | Packaging | 267 | paper | full |
| Apple | Fruits/Veg | 218 | organic | n/a | RiceBrick | Packaging | 510 | plastic | n/a |
| Banana | Fruits/Veg | 132 | organic | n/a | SamyangSpicyNoodles | Packaging | 92 | plastic | n/a |
| BellPepper | Fruits/Veg | 180 | organic | n/a | SanlinCoconutWater | Packaging | 354 | paper | full |
| Carrot | Fruits/Veg | 228 | organic | n/a | ShinRamyunCup | Packaging | 92 | paper | n/a |
| Corn | Fruits/Veg | 250 | organic | n/a | Snickers | Packaging | 52 | plastic | n/a |
| Cucumber | Fruits/Veg | 246 | organic | n/a | Spaghetti | Packaging | 506 | plastic | n/a |
| Orange | Fruits/Veg | 277 | organic | n/a | Sprite-500ml | Packaging | 529 | plastic | full |
| Pear | Fruits/Veg | 350 | organic | n/a | WatsonsSoda-330ml-B | Packaging | 345 | alum | full |
| Tomato | Fruits/Veg | 177 | organic | n/a | WatsonsSoda-330ml-R | Packaging | 345 | alum | full |
| GlassBowl | Kitchenware | 273 | glass | empty | Dumbbell | Sports | 1082 | metal | n/a |
| Mug | Kitchenware | 350 | ceramic | empty | KidsBasketball | Sports | 163 | rubber | n/a |
| Pot | Kitchenware | 446 | metal | empty | TennisBall | Sports | 57 | rub/felt | n/a |
| Clip | Office Supplies | 36 | metal | n/a | PaintBrush | Tools | 31 | wood | n/a |
| NarrowTape | Office Supplies | 135 | plastic | n/a | Screwdriver | Tools | 43 | met/plas | n/a |
| WideTape | Office Supplies | 266 | plastic | n/a | TapeMeasure | Tools | 84 | plas/met | n/a |
| 7Up-550ml | Packaging | 578 | plastic | full | WoodenStick | Tools | 93 | wood | n/a |
| CannedLunchMeat | Packaging | 372 | metal | full | ModelCar | Toys | 158 | plastic | n/a |
| CestbonWater | Packaging | 591 | plastic | full | RubiksCube | Toys | 56 | plastic | n/a |
| CocaCola-330ml | Packaging | 355 | alum | full | | | | | |

*Table 8.* Demographic statistics of the 12 participants in our dataset. Note that all participants are right-hand dominant.

| ID | Gender | Age | Weight | Body Fat | Hand Length |
|---|---|---|---|---|---|
| p001 | F | 29 | 52 kg | 19% | 17.1 cm |
| p002 | M | 29 | 77 kg | 20% | 18.0 cm |
| p003 | F | 24 | 59 kg | 24% | 17.7 cm |
| p004 | F | 23 | 68 kg | 22% | 18.3 cm |
| p005 | M | 28 | 71 kg | 20% | 19.6 cm |
| p006 | F | 27 | 55 kg | 27% | 17.8 cm |
| p007 | M | 25 | 74 kg | 25% | 18.4 cm |
| p008 | M | 23 | 68 kg | 21% | 18.8 cm |
| p009 | F | 27 | 58 kg | 25% | 17.7 cm |
| p010 | M | 28 | 80 kg | 26% | 18.9 cm |
| p011 | F | 25 | 44 kg | 24% | 17.7 cm |
| p012 | M | 25 | 78 kg | 23% | 18.9 cm |

### A.2.2. PRESSURE NORMALIZATION AND SCALING

To handle the large dynamic range (typically $[0, 350]\,$N) and occasional signal spikes, we implement a robust normalization pipeline. The normalization process achieves robust scaling by setting $p_{\max} = 200\,$N, corresponding to the 99.9th percentile of training samples. Under this proto-col, raw readings are denoised by zeroing out values below a per-sensor noise threshold $p_{\min} = 5\,$N, clipped at the $p_{\max}$ upper bound, and linearly mapped to the $[0, 1]$ interval.

Unless otherwise noted, models are trained to predict this normalized representation. We define the contact state for each individual sensing element based on this local threshold; specifically, any taxel value below $p_{\min}$ is treated as non-contact. In the normalized space, this corresponds to a per-pixel contact threshold of $\tau = p_{\min}/p_{\max}$. This threshold $\tau$ is used consistently across all contact-based metrics to evaluate the precision of predicted contact patterns.

### A.2.3. LOG-SPACE DISCRETIZATION

Egocentric grasp pressures exhibit a heavy-tailed distribution, as illustrated in Figure 6(a): high responses are sparse and concentrated on contact points, while the vast majority of the spatiotemporal volume remains near zero. Consequently, uniform quantization leads to severe class imbalance and insufficient resolution in the critical low-force regime. To address this, we employ a logarithmic quantization scheme that operates in the shifted log-space

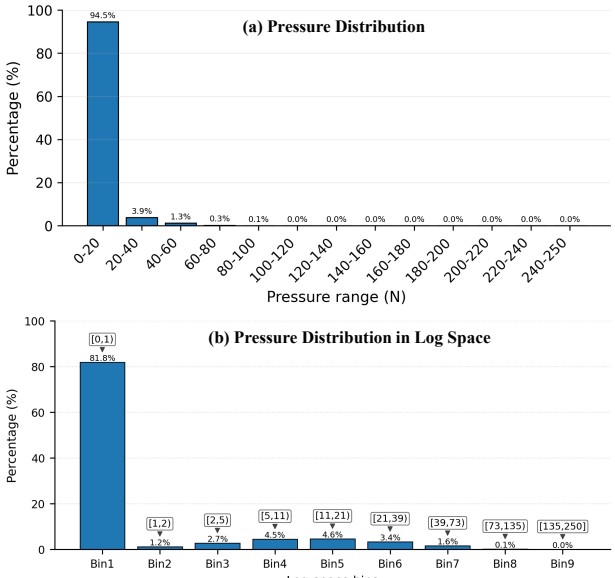

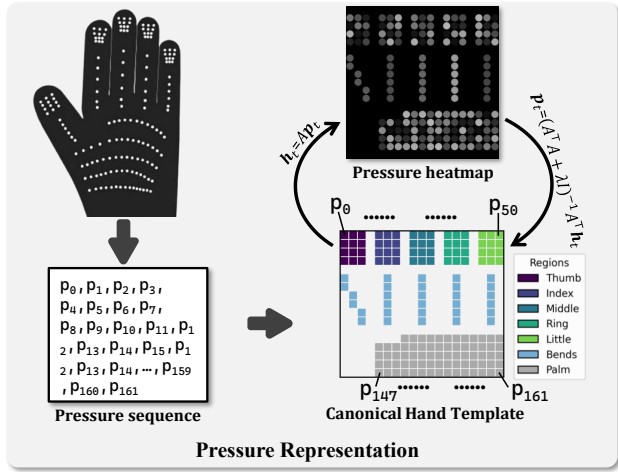

*Figure 7.* Pressure Representation and Bi-directional Conversion. We model the relationship between the sparse sensor sequence $\mathbf{p}_t$ and the dense pressure heatmap $\mathbf{h}_t$ via a canonical hand template. The forward linear operator $A$ diffuses sensor readings into a visual heatmap, while the inverse operation (right arrow) allows recovering discrete sensor values from visual predictions.

*Figure 6.* Pressure value distribution analysis. (a) The raw pressure distribution exhibits a significant long-tail property, where over 94% of values are concentrated in the near-zero range (0-20 N), causing severe class imbalance. (b) Log-space discretization strategy re-distributes the values into $K = 9$ bins with exponentially increasing widths, effectively preserving resolution for fine-grained interactions while covering the full dynamic range.

$\ln(p + 1)$, resulting in a more balanced distribution shown in Figure 6(b). This approach ensures higher resolution for small pressure values while covering the full range up to peak forces.

Let $K$ be the number of bins (e.g., $K = 9$) and $[v_{\min}, v_{\max}]$ be the pressure range (typically $v_{\min} = 0$). The bin edges $\{b_k\}_{k=0}^{K}$ are computed by linearly interpolating in the logarithmic domain and then mapping back to the linear domain:

$$
b_k = \text{round}\bigg( \exp\Big( \ln(v_{\min} + 1) + \\
\frac{k}{K}\big[\ln(v_{\max} + 1) - \ln(v_{\min} + 1)\big]\Big) - 1\bigg), \quad (5)
$$

for $k = 0, \ldots, K$. For a processed pressure value $p$, we assign a discrete label $y \in \{1, \ldots, K\}$ such that:

$$
b_{y-1} \le p < b_y. \quad (6)
$$

Note that the last bin is inclusive of the upper bound $v_{\max}$. This formulation effectively segments the pressure into $K$ levels, where the bin widths increase exponentially with pressure magnitude. This discretization strategy is specifically adopted by PressureVision (Grady et al., 2022) and EgoPressureFormer to map the raw continuous pressure into $K$ bins to mitigate learning difficulty.

### A.2.4. HAND PRESSURE HEATMAP CONSTRUCTION

We represent per-frame glove pressures as a hand pressure heatmap on a square canvas of size $S \times S$ (default $S = 256$). To bridge the modality gap between sparse sensor readings and dense visual cues, we formulate this conversion as a deterministic linear mapping derived from a canonical hand template, as illustrated in Figure 7. Specifically, we first establish the spatial domain by projecting the template onto the canvas via an aspect-ratio preserving affine transformation. Let $\mathcal{V}$ denote the set of valid foreground pixels on this fitted grid, with cardinality $|\mathcal{V}| = N_\mathcal{V}$. Given the projected coordinates $\{\mathbf{c}_m\}_{m=1}^{M}$ of the $M$ sensing anchors and the coordinate $\mathbf{u}_i$ for each pixel $i \in \mathcal{V}$, we precompute a fixed linear rendering operator in the form of a row-normalized matrix $A \in \mathbb{R}^{N_\mathcal{V} \times M}$. The elements of $A$ model the spatial diffusion of pressure from sensors to heatmap pixels using a row-normalized Gaussian kernel:

$$
A_{i,m} = \frac{\exp\big(-\|\mathbf{u}_i - \mathbf{c}_m\|_2^2/(2\sigma^2)\big)}{\sum_{m'=1}^{M} \exp\big(-\|\mathbf{u}_i - \mathbf{c}_{m'}\|_2^2/(2\sigma^2)\big) + \varepsilon}, \quad (7)
$$

where $\sigma$ controls the smoothness and $\varepsilon$ ensures numerical stability. This formulation allows us to treat the rendering process as a highly efficient matrix-vector multiplication, where $A$ depends only on the template geometry and remains constant across all frames.

### A.2.5. BI-DIRECTIONAL CONVERSION BETWEEN SEQUENCE AND HEATMAP

The linear operator $A$ enables efficient conversion between the sensor and visual domains.

*Forward Rendering:* Given a pressure vector $\mathbf{p}_t \in \mathbb{R}^M$, the heatmap intensities on the valid pixel set $\mathcal{V}$ are computed via matrix multiplication:

$$\mathbf{h}_t = A\mathbf{p}_t \in \mathbb{R}^{N_{\mathcal{V}}}. \tag{8}$$

These values are then populated into the canonical spatial domain to form the full heatmap $\mathbf{H}_t \in \mathbb{R}^{S \times S}$, with background regions zero-filled.

*Inverse Recovery:* Since the forward projection involves spatial smoothing and is non-invertible via simple sampling, we formulate the inverse problem as a Ridge regression. We search for the optimal latent pressure vector $\mathbf{p}$ that minimizes the reconstruction error given a heatmap observation $\mathbf{h}_t$:

$$\widehat{\mathbf{p}}_t = \arg \min_{\mathbf{p} \in \mathbb{R}^M} \|A\mathbf{p} - \mathbf{h}_t\|_2^2 + \lambda\|\mathbf{p}\|_2^2 = (A^\top A + \lambda I)^{-1} A^\top \mathbf{h}_t, \tag{9}$$

where $\lambda$ is a regularization hyperparameter. The result is optionally rectified ($\max(0, \widehat{\mathbf{p}}_t)$) to ensure physical plausibility.

# B. Method Implementation Details

## B.1. Backbone Architectures

### B.1.1. STABLE VIDEO DIFFUSION

Stable Video Diffusion (Blattmann et al., 2023) is adapted as a foundational generative backbone to leverage its robust temporal modeling capabilities. SVD extends the standard 2D Latent Diffusion Model (Rombach et al., 2022) by inflating the 2D U-Net (Ronneberger et al., 2015) architecture into a 3D counterpart. Specifically, it inserts temporal convolution and attention layers after every spatial convolution and attention block within the pre-trained 2D network. This design allows the model to learn temporal dependencies and motion dynamics across video frames while retaining the strong spatial priors of the original image-based model. In the context of our EgoPressureDiff framework, this architecture provides a powerful reference for modeling the continuous temporal evolution of tactile signals from visual cues, ensuring that the generated pressure sequences maintain high temporal coherence consistent with the input egocentric video. Moreover, benefiting from large-scale pre-training on diverse real-world footage, the SVD backbone injects rich world knowledge priors into our framework. This empowers EgoPressureDiff with an implicit understanding of object physics, allowing the model to infer potential physical attributes, such as material stiffness and weight distribution, directly from the visual appearance of the grasped objects.

### B.1.2. TIMESFORMER

TimeSformer (Bertasius et al., 2021) is employed as a visual encoder to extract rich spatiotemporal features from the input egocentric video clips. Unlike traditional 3D convolutional neural networks, TimeSformer adapts the Vision Transformer (Dosovitskiy, 2020) architecture to the video domain by treating the video as a sequence of frame-level patches. Its core contribution is the Divided Space-Time Attention mechanism, which sequentially applies temporal attention and spatial attention. This factorized design significantly reduces computational complexity while effectively capturing both long-range temporal dependencies and spatial details. For our task, TimeSformer serves as an efficient backbone to encode the complex hand-object interactions in the EgoTactile dataset, providing the necessary spatiotemporal embeddings for downstream pressure estimation.

## B.2. EgoPressureFormer Details

EgoPressureFormer is a discriminative sequence predictor that maps an egocentric clip to per-sensor pressure sequences. Instead of predicting pixel-aligned pressure maps, it directly predicts the pressure of the $M$ glove sensors using anatomy-aware queries and a frame-level contact gate to mitigate sparsity in contact frames. Unless otherwise stated, EgoPressureFormer predicts discrete pressure bins and can be deterministically converted to continuous pressure by bin expectation.

### B.2.1. NETWORK ARCHITECTURE

Given a clip of $T$ RGB frames, we feed it into a TimeSformer video transformer to obtain spatiotemporal token features at each time step. Concretely, for each sampled frame $t$, the backbone produces a set of visual tokens. The temporal modeling is handled internally by the TimeSformer attention mechanism. On top of these features, we attach a lightweight decoder with two heads:

*(1) Per-sensor pressure head.* We maintain $M{=}162$ learnable sensor embeddings, one for each glove sensor. For each frame $t$, we use a small cross-attention block where these sensor embeddings attend to the visual tokens of that frame, producing $M$ sensor-specific features that summarize the video evidence relevant to each sensor. These features are then passed through a shared MLP to output per-sensor logits over $K$ pressure bins for each frame, resulting in a tensor of shape $T \times M \times K$. This decoding strategy avoids requiring explicit pixel-to-sensor alignment, as the model learns to retrieve the most relevant visual evidence for each sensor through attention, which is particularly important under egocentric occlusions.

*(2) Frame-level contact head.* In parallel, we derive a frame

representation by mean-pooling the visual tokens at each time step, and apply an MLP to predict a single contact logit per frame, resulting in a length-$T$ sequence. This contact score is used for gating supervision during training.

### B.2.2. TRAINING OBJECTIVE

We train EgoPressureFormer with a multi-task objective that combines frame-level contact classification and per-sensor pressure-bin classification, where the pressure loss is applied only on contact frames.

Denote by $a_t \in \{0, 1\}$ the ground-truth contact indicator at time step $t$, and by $\widehat{a}_t \in \mathbb{R}$ the corresponding predicted contact logit. For the pressure supervision, let $y_{t,m} \in \mathbb{R}^K$ be the ground-truth discrete pressure-bin label of sensor $m$ at time step $t$, and let $\widehat{y}_{t,m} \in \mathbb{R}^K$ be the predicted logits. We optimize the weighted sum

$$\mathcal{L} = \lambda_{\text{cls}}\mathcal{L}_{\text{contact}} + \lambda_{\text{press}}\mathcal{L}_{\text{press}}, \quad (10)$$

where $\lambda_{\text{cls}}$ and $\lambda_{\text{press}}$ weight the contact and pressure terms, respectively (default $\lambda_{\text{cls}} = 5$ and $\lambda_{\text{press}} = 1$). The contact loss is a binary classification loss:

$$\mathcal{L}_{\text{contact}} = \frac{1}{T} \sum_{t=1}^{T} \ell_{\text{BCE}}(\widehat{a}_t, a_t), \quad (11)$$

where $\ell_{\text{BCE}}(\cdot, \cdot)$ is the binary cross-entropy between a logit and a binary label. The pressure loss is a masked cross-entropy over bins:

$$\mathcal{L}_{\text{press}} = \frac{1}{\sum_{t=1}^{T} a_t + \varepsilon} \sum_{t=1}^{T} a_t \cdot \left( \frac{1}{M} \sum_{m=1}^{M} \ell_{\text{CE}}(\widehat{y}_{t,m}, y_{t,m}) \right), \quad (12)$$

where $\ell_{\text{CE}}(\cdot, \cdot)$ is the standard $K$-way cross-entropy between a logit vector and a class index, and $\varepsilon$ is a small constant to avoid division by zero when a clip contains no contact frames. This gated formulation focuses pressure supervision on informative contact frames, while non-contact frames are primarily learned through the contact objective.

### B.3. Heatmap post-processing for EgoPressureDiff

To rigorously bridge the gap between the generative diffusion outputs and the physical constraints of the tactile sensors, we implement a post-processing pipeline. Although EgoPressureDiff generates perceptually coherent patterns, the raw output may contain noise that violates the inverse recovery process defined in Eq. 9. We first standardize the intensity domain by converting the generated heatmap to a single-channel representation via channel-wise averaging, followed by robust percentile-based normalization restricted to the valid hand region $\mathcal{V}$, yielding the normalized heatmap $\hat{\mathbf{H}}_{norm}$. To enforce physical consistency, we project the heatmap onto the manifold of valid pressure distributions

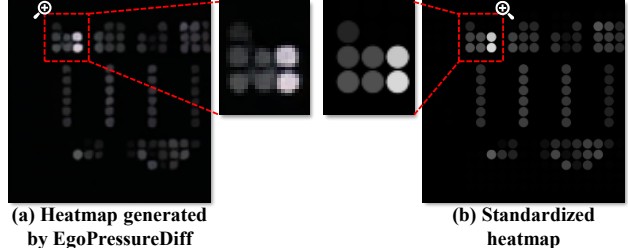

| (a) Heatmap generated by EgoPressureDiff | (b) Standardized heatmap |

*Figure 8.* Visualization of the Heatmap Standardization Process. We compare one of the lowest-quality raw heatmaps generated by EgoPressureDiff (Left) with the physically standardized heatmap (Right). The pipeline filters out generative noise and enforces anatomical consistency by re-projecting the prediction.

defined by the sensor operator $A$. An intermediate pressure vector $\hat{\mathbf{p}}' \in \mathbb{R}^M$ is estimated by aggregating localized intensities around each sensor's projected coordinate $\mathbf{c}_m$:

$$\hat{\mathbf{p}}'_m = \frac{1}{|\mathcal{N}(\mathbf{c}_m, r)|} \sum_{\mathbf{u} \in \mathcal{N}(\mathbf{c}_m, r)} \hat{\mathbf{H}}_{norm}(\mathbf{u}), \quad (13)$$

where $\mathcal{N}(\mathbf{c}_m, r)$ denotes the set of pixels within a disk neighborhood of radius $r$ centered at $\mathbf{c}_m$ ($r = 3$ by default). We explicitly re-render the standardized heatmap vector as $\hat{\mathbf{h}}_{std} = A\hat{\mathbf{p}}'$. These values are subsequently mapped onto the canonical spatial domain to form the full heatmap $\hat{\mathbf{H}}_{std}$, with background regions zero-filled. As demonstrated in Figure 8, this re-projection acts as a physical filter, suppressing generative hallucinations and high-frequency noise. Finally, the pressure sequence is recovered from this clean observation $\hat{\mathbf{H}}_{std}$ using a robust regression solver, ensuring the final predictions are both visually sharp and physically plausible.

## C. Training Settings and Evaluation Metrics

### C.1. Training Details & Hyperparameters

#### C.1.1. EGOPRESSUREFORMER

For the discriminative baseline, we employ the TimeSformer architecture with divided space-time attention. The model is trained on the EgoTactile dataset using 2 NVIDIA A800 (80GB) GPUs. The input video clips consist of 16 frames sampled with a temporal stride of 5, and each frame is resized and cropped to a spatial resolution of $256 \times 256$. Similar to the previous baseline, all parameters of the TimeSformer network are updated during training. We set the total batch size to 16. The model is optimized using SGD with a momentum of 0.9 and a base learning rate of 0.005. We employ a step-wise learning rate decay policy and train the model for a total of 30 epochs.

### C.1.2. EGOPRESSUREDIFF

EgoPressureDiff is initialized with the pre-trained weights of the Stable Video Diffusion img2vid-xt checkpoint. To incorporate explicit spatial guidance, we employ Grounded-SAM (Ren et al., 2024) to extract binary hand masks from the input frames, which serve as hint inputs. During training, we resize all input egocentric video frames to a spatial resolution of $256 \times 256$. Each training sample consists of a clip of 16 frames, sampled at a frame rate of 5 fps. The model is fine-tuned on the EgoTactile dataset using 2 NVIDIA A800 (80GB) GPUs. We optimize the 3D U-Net backbone and the Mask Encoder, while keeping all other pre-trained components frozen. We set the per-device batch size to 2, resulting in a global batch size of 4. Optimization is performed using the AdamW optimizer with a fixed learning rate of $1 \times 10^{-5}$, following a linear warmup phase of 500 steps. We employ mixed-precision training to optimize memory usage and train the model for a total of 40k steps. The text prompt follows a structured template to encode object and subject attributes: *"This video shows the action of picking up {obj_name}. The weight of {obj_name} is {weight}, and its surface material is {material}. The person performing the action is {gender}, {age} years of age, weighing {subj_weight} and having {bodyfat} body fat."*

### C.1.3. PRESSUREVISION

For PressureVision, to ensure fair comparison, we use the same training split as other baselines. We discretize continuous pressure into $K=9$ log-space bins to construct heatmap targets, mitigating the heavy-tailed pressure distribution. Using $256\times256$ inputs, we perform full fine-tuning where all model parameters are updated for 20 epochs with a batch size of 16 and an initial learning rate of 0.001, decayed by 0.1 at epoch 10.

### C.2. Part-wise Center-of-Pressure (CoP) Error

Standard metrics such as Volumetric IoU and MAE focus primarily on the magnitude and global overlap of pressure signals. However, they lack the granularity to evaluate whether the predicted pressure concentrates at the correct locations within the hand. To quantify spatial localization performance under occlusion and multi-contact grasps, we introduce the part-wise CoP error, as visually illustrated in Figure 9. This metric is computed separately on each fingertip and the palm, and then averaged.

Let $\mathbf{c}_m \in \mathbb{R}^2$ denote the 2D fitted-grid coordinate of sensor $m$, and let $\{\mathcal{R}_r\}_{r=1}^6$ be the index sets corresponding to the five fingertips and the palm. For a specific anatomical part $r$ at frame $t$, we determine valid contact using an activation density criterion. Specifically, a part is deemed in contact only if more than 20% of its constituent sensors register a

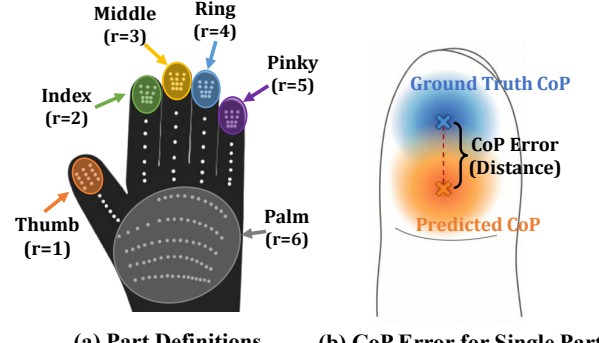

**(a) Part Definitions**  **(b) CoP Error for Single Part**

*Figure 9.* Illustration of the Part-wise Center-of-Pressure (CoP) Error metric. We compute the Euclidean distance between the pressure-weighted centroids of the predicted and ground-truth heatmaps for each anatomical part.

normalized pressure value above the per-sensor threshold $\tau$, which is detailed in Appendix A.2.2. We compute the CoP only when this significant contact condition is met in either the ground truth or the prediction.

The part-wise CoP is defined as the pressure-weighted centroid:

$$\mathbf{g}_t^{(r)} = \frac{\sum\limits_{m \in \mathcal{R}_r} p_t[m]\, \mathbf{c}_m}{\sum\limits_{m \in \mathcal{R}_r} p_t[m] + \varepsilon}, \quad \widehat{\mathbf{g}}_t^{(r)} = \frac{\sum\limits_{m \in \mathcal{R}_r} \widehat{p}_t[m]\, \mathbf{c}_m}{\sum\limits_{m \in \mathcal{R}_r} \widehat{p}_t[m] + \varepsilon}, \quad (14)$$

where $\varepsilon$ is a small constant ($1e^{-6}$) added for numerical stability. The CoP error for part $r$ is then calculated as the average Euclidean distance over all valid contact frames:

$$\mathrm{Err}_{\mathrm{CoP}}^{(r)} = \frac{1}{|\mathcal{T}_r|} \sum_{t \in \mathcal{T}_r} \left\| \widehat{\mathbf{g}}_t^{(r)} - \mathbf{g}_t^{(r)} \right\|_2,$$

where $\mathcal{T}_r = \{t \mid \mathrm{Ratio}(p_t, \mathcal{R}_r) > 0.2 \vee \mathrm{Ratio}(\widehat{p}_t, \mathcal{R}_r) > 0.2\},$

$$\mathrm{Ratio}(p, \mathcal{R}) = \frac{\sum_{m \in \mathcal{R}} \mathbb{I}(p[m] > \tau)}{|\mathcal{R}|}.$$

$$(15)$$

Here, $\mathbb{I}(\cdot)$ denotes the indicator function. Finally, the system-level metric is reported as the mean error over all six anatomical parts:

$$\mathrm{Err}_{\mathrm{CoP}} = \frac{1}{6} \sum_{r=1}^{6} \mathrm{Err}_{\mathrm{CoP}}^{(r)}. \quad (16)$$

This metric serves as a complementary measure to PressureVision's overlap-based scores, specifically targeting the model's ability to localize precise contact points on the hand surface.

## D. Extended Related Work

### D.1. Visual-Tactile Datasets

Understanding the fine-grained physical dynamics of hand-object interaction is a core challenge in computer vision

*Table 9.* Comparison of representative visual-tactile and visual pressure estimation datasets with EgoTactile. We compare key characteristics including viewpoint, interaction surface, pressure coverage, scale, and collection environment. While most datasets typically provide only object names, ours includes granular participant metadata (age, body weight, body fat rate, gender, dominant hand) and detailed object properties (material types, weight, filled states).

| Dataset | Viewpoint | Interaction Surface | Pressure Coverage | Scale | Participants | Objects | Subj. Attr. | Obj. Attr. |
|---|---|---|---|---|---|---|---|---|
| *Full-hand Tactile Signatures* | | | | | | | | |
| STAG (Sundaram et al., 2019) | Exo | 3D objects | Full hand (548 taxels) | 13k frames | 1 | 26 | ✗ | ✓ |
| ADL-21 (Cepriá-Bernal & Pérez-González, 2021) | – | 3D objects | Full hand (361 taxels) | 323k frames | 22 | – | ✓ | ✗ |
| PiMForce (Seo et al., 2024) | – | 3D objects | Full hand (9 taxels) | 1,242k frames | 21 | – | ✗ | ✗ |
| *Full-hand activity datasets* | | | | | | | | |
| ActionSense (DelPreto et al., 2022) | Ego + Exo | 3D objects (kitchen) | Full hand (682 taxels) | 1.4M frames | 10 | 21 | ✓ | ✓ |
| OPENTOUCH (Song et al., 2025) | Ego | 3D objects | Full hand (169 taxels) | 550k frames | – | 8,000 | ✗ | ✗ |
| *Visual-tactile pretraining/action* | | | | | | | | |
| M$^2$VTP (Liu et al., 2024) | Ego | 3D object (bottle-cap) | Full hand (20 taxels) | 3k frames | 1 | 20 | ✗ | ✗ |
| VTDexManip (Liu et al., 2025) | Ego | 3D objects (10 tasks) | Full hand (20 taxels) | 565k frames | 5 | 182 | ✗ | ✗ |
| HumanoidVTA (Kwon et al., 2025) | Robot Ego | 3D objects | Full hand (1,062 taxels) | 102k frames | – | 2 | ✗ | ✗ |
| *Visual pressure estimation* | | | | | | | | |
| PressureVisionDB (Grady et al., 2022) | Exo | Planar surface | – | 3.0M frames | 36 | – | ✗ | ✗ |
| ContactLabelDB (Grady et al., 2024) | Exo | Diverse surfaces | – | 2.9M frames | 51 | 106 | ✗ | ✗ |
| EgoPressure (Zhao et al., 2025) | Ego+Exo | Planar surface | – | 4.3M frames | 21 | – | ✓ | ✗ |
| **EgoTactile (Ours)** | **Ego** | **3D objects** | **Full hand (162 taxels)** | **319k frames** | **12** | **63** | ✓ | ✓ |

and robotics. To address this, a growing body of research has focused on curating visual-tactile datasets that bridge visual perception with physical contact signals. Existing benchmarks can be broadly categorized by their data capture modalities and target applications, ranging from visual pressure estimation and egocentric activity recognition to multimodal representation learning for manipulation. Here we review these developments to contextualize the unique contribution of our work in egocentric full-hand pressure prediction, and summarize key characteristics in Table 9.

*Visual pressure estimation datasets.* PressureVision (Grady et al., 2022) collects RGB videos of bare hands pressing an instrumented planar pressure surface and trains models to infer a dense pressure image from a single RGB frame. The dataset covers 36 participants but is limited to tabletop-like contact scenarios. EgoPressure (Zhao et al., 2025) extends this setting by introducing an egocentric head-mounted view and multi-view pose/mesh annotations, providing 5 hours of interactions from 21 participants on a Sensel Morph touch-pad, and projecting pressure onto the hand mesh as UV textures. PressureVision++ (Grady et al., 2024) further increases diversity by capturing fingertip pressure on natural objects/surfaces, but its supervision focuses on fingertips rather than full-hand contact. These datasets either constrain interaction to planar surfaces or provide only partial pressure coverage, and they do not target full-hand force distribution during 3D object grasping.

*Egocentric full-hand touch and activity datasets.* OPEN-TOUCH (Song et al., 2025) is the first in-the-wild egocentric dataset aligning RGB video with full-hand tactile maps and 3D hand pose, enabling retrieval/classification benchmarks for contact-rich interactions. However, since it mainly collects data in uncontrolled scenarios, it is less suitable for

controlled variable analysis during the model learning process. ActionSense (DelPreto et al., 2022) provides a broader multimodal activity capture in a kitchen setting (wearables + external cameras), including tactile, gaze, EMG, and audio signals, but it is not designed for pressure prediction on grasp objects and does not emphasize dense full-hand pressure supervision for grasping.

*Visual-tactile pretraining and dexterous manipulation datasets.* Recent work explores visual-tactile representation learning for manipulation. M$^2$VTP (Liu et al., 2024) collects egocentric Hololens2 RGB with a low-cost glove of 20 binary contact sensors, forming 120 sequences of bottle-cap turning for masked visual–tactile pretraining. VT-DexManip (Liu et al., 2025) scales this idea to 2,032 human manipulation sequences over 10 daily tasks and 182 objects (565k frame pairs) using egocentric vision and sparse tactile signals (often binarized for sim-to-real). In robotics, humanoid visual-tactile-action datasets have also emerged (Kwon et al., 2025), but they focus on tele-operated robot embodiments rather than human full-hand pressure prediction from egocentric video.

*Full-hand tactile gloves and force-related human datasets.* Beyond vision-centric supervision, tactile gloves enable direct measurement of distributed contact forces during grasping. STAG (Sundaram et al., 2019) records high-resolution full-hand tactile maps during object interaction, demonstrating that tactile signatures encode object identity and weight-related cues. Complementary to STAG, ADL-21 (Cepriá-Bernal & Pérez-González, 2021) releases an open dataset of full-hand tactile signatures for 21 activities of daily living repeated by 22 subjects, captured with a commercial Tekscan Grip System sewn onto a glove, and the authors analyze task/subject effects and inter-region force

sharing patterns. PiMForce (Seo et al., 2024) further integrates whole-hand pressure with physiology and posture (pressure glove + sEMG + 3D hand posture), but it is not framed as an egocentric visual pressure prediction benchmark on diverse 3D objects.

In contrast to prior datasets, EgoTactile provides large-scale paired hand pressure and egocentric video data captured in controlled scenarios involving the grasping of diverse daily objects. Notably, our dataset includes numerous scenarios with hand occlusions, facilitating rigorous scientific experimentation and analysis. Furthermore, we provide detailed physical attributes and subject metadata, enabling in-depth analysis of the correlations between pressure magnitude and various physical factors.

## E. Additional Experiments and Analysis

### E.1. Consistency of the Bare-Hand Protocol

The bare-hand subset is designed as a weakly paired transfer benchmark: the visible hand is bare, while a synchronized off-camera gloved hand provides pressure labels. To quantify the consistency of this protocol, we conduct a dual-glove validation study that simulates the bare-hand collection procedure. Both hands wear pressure gloves. One hand is visible to the egocentric camera, while the other grasps the same object outside the camera view. We collect 3 subjects $\times$ 5 held-out objects $\times$ 10 repetitions, resulting in 150 clips. We compare the pressure streams from the two hands directly, rather than comparing model predictions with ground truth.

*Table 10.* Dual-glove consistency under the bare-hand protocol. Temporal gaps (ms) are measured on the synchronized pressure streams.

| Object | $\Delta$**Onset**↓ | $\Delta$**Release**↓ | $\Delta$**Duration**↓ | **C-IoU**↑ | **CoP**↓ | **MAE**↓ |
|---|---|---|---|---|---|---|
| Apple | 101 | 116 | 149 | 82.6 | 2.4 | 5.1 |
| Cola330ml | 94 | 108 | 136 | 84.1 | 2.2 | 4.7 |
| Corn | 126 | 139 | 181 | 79.8 | 2.8 | 6.3 |
| Dumbbell | 87 | 96 | 129 | 86.3 | 2.0 | 4.2 |
| TennisBall | 118 | 131 | 174 | 80.5 | 2.7 | 5.8 |
| Overall | 105±31 | 118±36 | 154±42 | 82.7±5.6 | 2.4±0.7 | 5.2±1.6 |

As shown in Table 10, the two hands exhibit small timing gaps and high contact agreement. The average contact-onset and release gaps are 105 ms and 118 ms, respectively, corresponding to less than two frames on the 15 Hz synchronized timeline. This indicates that the weak pairing is not exact, but the resulting pressure labels remain sufficiently consistent for evaluating transfer to natural bare-hand inputs. We further measure cross-trial repeatability with 1 subject, 3 objects, and 30 trials per object. For each trial, we compare its mean contact pressure map against the average of the other 29 trials of the same object.

*Table 11.* Cross-trial repeatability under the bare-hand protocol.

| Object | $\Delta$**Onset(ms)**↓ | **C-IoU(%)**↑ | **MAE(N)**↓ |
|---|---|---|---|
| Apple | 148±39 | 86.4±4.2 | 4.6±1.0 |
| CocaCola330ml | 131±34 | 88.7±3.8 | 4.0±0.9 |
| Dumbbell | 156±43 | 85.1±4.6 | 4.9±1.1 |
| Overall | 145±39 | 86.7±4.2 | 4.5±1.0 |

### E.2. Effect of Object and Subject Attributes

We further analyze the contribution of different text attributes in EgoPressureDiff. The full text prompt contains both object-side attributes, such as object name, material, weight, and load state, and subject-side attributes, such as gender, age, body weight, and body fat rate. Table 12 shows that object attributes are the dominant source of physical conditioning, while subject attributes provide complementary individual priors, especially under the Subject-Held-Out protocol.

*Table 12.* Effect of object and subject information in text conditioning.

| Variant | Object-Held-Out | | Subject-Held-Out | |
|---|---|---|---|---|
| | V-IoU(%)↑ | MAE(N)↓ | V-IoU(%)↑ | MAE(N)↓ |
| Full text conditioning | **38.9** | **3.4** | **34.0** | **5.8** |
| w/o subject info | 38.2 | 3.8 | 30.9 | 6.6 |
| w/o object info | 29.9 | 4.9 | 28.4 | 6.8 |
| w/o text | 30.3 | 5.1 | 28.0 | 7.1 |

Removing subject information causes only a small drop under Object-Held-Out evaluation, suggesting that object attributes dominate force magnitude estimation when object-side physical cues are available. The drop is larger under Subject-Held-Out evaluation, indicating that subject attributes help calibrate individual grasping tendencies and force scales for unseen participants. The overall gain from subject attributes is still limited by the current number of participants and their demographic range, and we expect this effect to become more pronounced with larger-scale subject diversity.

### E.3. Efficiency Analysis

**Resolution trade-off.** We use $256 \times 256$ resolution for RGB frames, masks, and pressure heatmaps in both training and inference. Since the final pressure representation is mapped to 162 taxels, the target signal is relatively low-dimensional, and the main challenge comes from occlusion and visual-physical ambiguity rather than fine image details. Table 13 reports the accuracy-speed trade-off under different resolutions. The $128 \times 128$ setting is faster but degrades contact localization and pressure magnitude estimation. In contrast, $512 \times 512$ brings only marginal gains while substantially increasing computational cost. We therefore adopt $256 \times 256$ as a balanced setting.

*Table 13.* Performance and speed under different RGB, mask, and pressure-map resolutions on the Object-Held-Out split.

| Resolution | C-IoU(%)↑ | V-IoU(%)↑ | MAE(N)↓ | CoP↓ | FPS↑ |
|---|---|---|---|---|---|
| 128×128 | 51.8 | 35.4 | 3.9 | 3.6 | 5.0 |
| 256×256 | 56.3 | 38.9 | 3.4 | 3.1 | 2.8 |
| 512×512 | **56.9** | **39.4** | **3.3** | **3.0** | 0.9 |

**Latent consistency distillation.** To accelerate EgoPressureDiff, we distill the original 25-step model into few-step latent consistency models. We keep the same task setting, 256×256 resolution, 16-frame clips, and Object-Held-Out split. During distillation, the VAE and CLIP encoders are frozen, while the 3D U-Net, Mask Encoder, and PIFR-related conditioning modules are distilled.

*Table 14.* Accuracy and speed trade-off after latent consistency distillation.

| Method | Steps | C-IoU(%)↑ | V-IoU(%)↑ | MAE(N)↓ | FPS↑ |
|---|---|---|---|---|---|
| EgoPressureDiff | 25 | **56.3** | **38.9** | **3.4** | 2.8 |
| LCM student | 8 | 54.9 | 37.3 | 3.7 | 6.9 |
| LCM student | 4 | 52.1 | 35.2 | 4.4 | **10.3** |

The 8-step student improves throughput by about 2.5× with only a small performance drop, while the 4-step student reaches 10.3 FPS with a larger degradation. This suggests that consistency distillation is a practical acceleration strategy for the current SVD-based architecture. Flow matching remains a promising future direction, but would require replacing the current denoising objective and redesigning the conditioning interfaces.

### E.4. Decoupling Physics from Appearance

A unique advantage of EgoPressureDiff is its ability to utilize text metadata to resolve visual ambiguities. We investigate this capability through a counterfactual prompting experiment using the model trained under the Object-Held-Out protocol on the gloved-hand set. We select an unseen object, a 330ml Coca-Cola can, and perform inference on the same video clip while varying the weight attribute within the input prompt. The default prompt provides comprehensive context: *"This video shows the action of picking up a Coca-Cola. The weight of the CocaCola is 355g, and its surface material is aluminum. The person performing the action is male, 25 years of age, weighing 74kg and having 25% body fat"*. We specifically test three explicit weight values (1g, 355g, 1000g) alongside a setting where weight information is omitted to observe the model's response. As visualized in Figure 11, altering the text prompt systematically shifts the predicted pressure distribution. For explicit weights, the distinction is evident: the 1000g prompt induces significantly higher pressure values, whereas the 1g prompt yields minimal activation. Interestingly, when weight information is omitted, the model still outputs reasonable pressure mag-

nitudes and accurate contact locations. We attribute this robustness to the rich world knowledge priors encapsulated in the pre-trained SVD backbone (Blattmann et al., 2023), which enables plausible estimation even in the absence of explicit physical constraints. This confirms that the PIFR layer successfully modulates diffusion features to bridge the data gap of invisible physical attributes, effectively resolving ambiguities that are inherently ill-posed for pure vision-based baselines.

### E.5. Qualitative Results in the Wild

While qualitative examples verify performance of EgoPressureDiff in controlled settings, applying the model to real-world scenarios requires resilience to significant domain shifts. To assess this capability, we evaluate EgoPressureDiff on a curated set of in-the-wild egocentric clips characterized by complex backgrounds and dynamic lighting conditions. In the absence of ground-truth sensor data for these sequences, we evaluate performance based on physical plausibility and temporal stability. As demonstrated in Figure 12 and Figure 13, EgoPressureDiff exhibits strong zero-shot robustness, consistently predicting physically plausible contact patterns despite environmental challenges. Furthermore, in Figure 14, we specifically test the model on objects unseen during training. Even in this challenging setting, the model generalizes effectively, inferring reasonable pressure distributions and contact geometries. A key factor enabling this success is the explicit conditioning on hand and object masks. These masks effectively guide the generative process to focus exclusively on the interaction dynamics between the hand and the target object, allowing the model to ignore distracting background textures.

### E.6. Additional Visualizations on Held-out Objects

To further substantiate the model's generalization capability, we provide extensive qualitative visualizations of EgoPressureDiff evaluated on unseen objects under the Object-Held-Out protocol. For the Gloved-hand set, we demonstrate performance across different egocentric viewpoints. Figure 15 illustrates the continuous pressure generation from a neck-mounted perspective for grasping a Corn (Left column) and a Tennis Ball (Right column). Complementing this, Figure 16 showcases generations from a head-mounted perspective for grasping a Dumbbell (Left column) and a CocaCola-330ml (Right column). These sequences confirm that our model maintains high temporal consistency and spatial accuracy across varying camera angles and object geometries. We also present results for the Bare-hand set under the same Object-Held-Out protocol to verify cross-domain transferability. As shown in Figure 17, we visualize the continuous pressure generation from a neck-mounted view for grasping an Apple (Left column) and a Dumbbell (Right column).

## E.7. Failure Cases

Despite its strong performance, EgoPressureDiff exhibits certain limitations characteristic of vision-based tactile estimation and generative modeling. We visualize two representative failure modes in Figure 10. As illustrated in the top row, the model struggles with precise contact estimation during grasp initiation and release. In the grasp initiation phase (Top-Left), the model fails to predict pressure despite visible contact. Conversely, during the release phase (Top-Right), it hallucinates lingering pressure after the hand has already detached from the object. We attribute these errors to severe egocentric self-occlusion, which obscures the subtle gap between the hand and the objects, making the exact determination of contact onset and offset visually ambiguous. The bottom row of Figure 10 highlights a limitation inherent to diffusion-based architectures, namely temporal instability. Although the video input depicts a continuous, stable grasp, the predicted pressure distribution within the highlighted region (red dashed box) exhibits noticeable flickering across adjacent frames. This artifact arises from the stochastic sampling process of the diffusion model, where independent noise initialization can lead to inter-frame variance in the generated heatmaps. Future work could address this by incorporating stronger temporal consistency constraints or consistency distillation techniques.

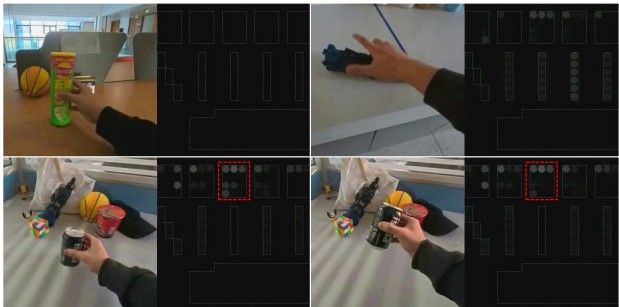

*Figure 10.* Failure Cases. Top Row: Ambiguity in contact transitions due to occlusion. The model yields a false negative during grasp initiation (Left) and a false positive during release (Right). Bottom Row: Temporal inconsistency in diffusion generation. Despite a continuous grasping action, the pressure magnitude on the fingers (red dashed box) fluctuates between adjacent frames, exhibiting the "flickering" artifact inherent to stochastic generative models.

## E.8. Generalization to Unseen Interaction Patterns

The main EgoTactile collection asks participants to perform natural grasps rather than scripted grasp taxonomies. Although this captures habitual everyday strategies, it may still lead to relatively common grasp configurations. To stress-test generalization, we construct an unseen interaction-pattern test set with 40 gloved-hand clips on 3 held-out objects. The interactions include palm press and roll, palm support, hook lift, toss-and-catch, and palm-base support. Models are evaluated without retraining.

*Table 15.* Generalization to unseen poses and interaction patterns. "Natural" denotes the controlled Object-Held-Out benchmark.

| Method | Setting | C-IoU(%)↑ | V-IoU(%)↑ | MAE(N)↓ | CoP↓ |
|---|---|---|---|---|---|
| PressureVision | Natural | 24.5 | 16.8 | 9.2 | 12.5 |
| | Unseen | 9.8 | 5.9 | 15.8 | 17.4 |
| | Δ | -14.7 | -10.9 | +6.6 | +4.9 |
| EgoPressureFormer | Natural | 36.8 | 26.5 | 6.2 | 7.5 |
| | Unseen | 16.7 | 10.8 | 10.9 | 13.2 |
| | Δ | -20.1 | -15.7 | +4.7 | +5.7 |
| EgoPressureDiff | Natural | 56.3 | 38.9 | 3.4 | 3.1 |
| | Unseen | **31.8** | **21.4** | **7.1** | **7.8** |
| | Δ | -24.5 | -17.5 | +3.7 | +4.7 |

All methods degrade clearly under rare interaction patterns. EgoPressureDiff remains the best-performing method, but the drop indicates that RGB-only conditioning is insufficient for fully modeling unfamiliar hand-object geometries. We attribute this limitation to the lack of explicit detailed hand pose and object 6D pose conditioning. This motivates future work on incorporating stronger geometric priors and action-level interaction modeling.

| Input Frame | "1g" | "355g" | "1000g" | omitted |
|---|---|---|---|---|

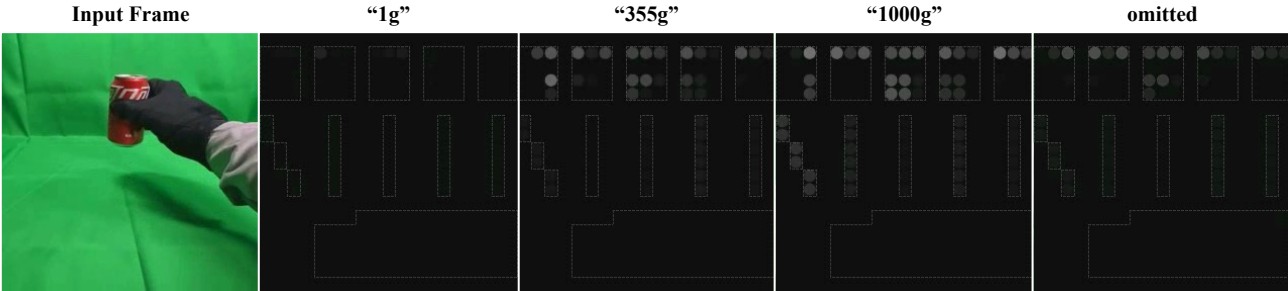

*Figure 11.* Counterfactual Prompting Analysis. Explicitly manipulating the weight attribute reveals the model's ability to decouple physics from appearance. The "1000g" prompt induces significantly higher pressure intensities compared to the "1g" prompt, which yields minimal activation. Notably, in the omitted case (Right) where no weight is specified, the model still generates a physically plausible pressure distribution and contact geometry. For clarity, dashed hand outlines are overlaid on the heatmaps to indicate the hand contour.

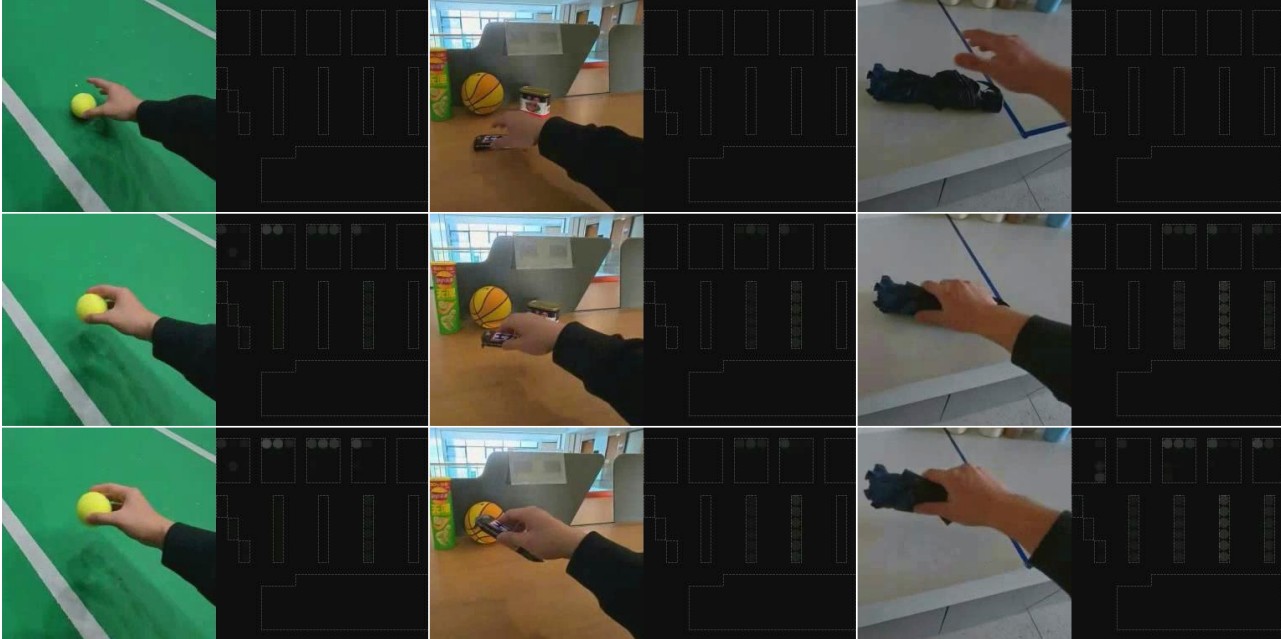

*Figure 12.* Qualitative robustness in the wild (1/3). We visualize the predictions of EgoPressureDiff on unconstrained real-world clips. Despite challenges like complex backgrounds, motion blur, and dynamic lighting, our model generates spatially precise and physically plausible pressure heatmaps. This verifies that the explicit mask conditioning effectively filters out environmental noise, enabling robust generalization beyond the laboratory setting.

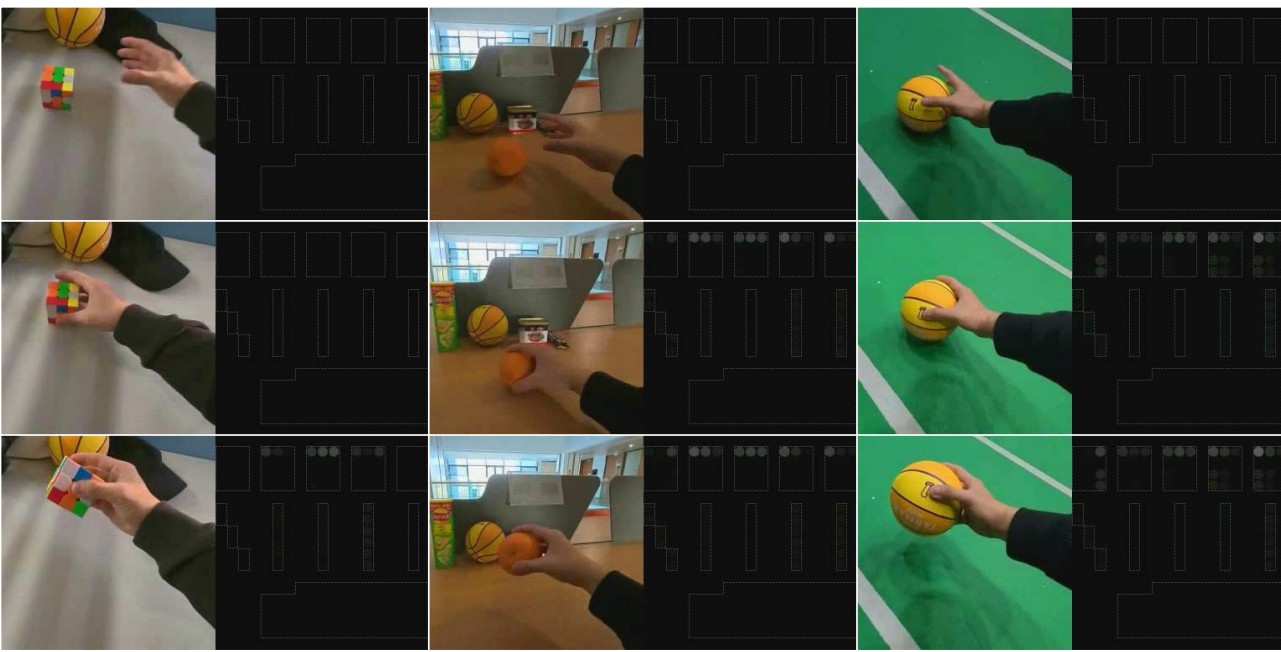

*Figure 13.* Qualitative robustness in the wild (2/3). We visualize the predictions of EgoPressureDiff on unconstrained real-world clips. Despite challenges like complex backgrounds, motion blur, and dynamic lighting, our model generates spatially precise and physically plausible pressure heatmaps. This verifies that the explicit mask conditioning effectively filters out environmental noise, enabling robust generalization beyond the laboratory setting.

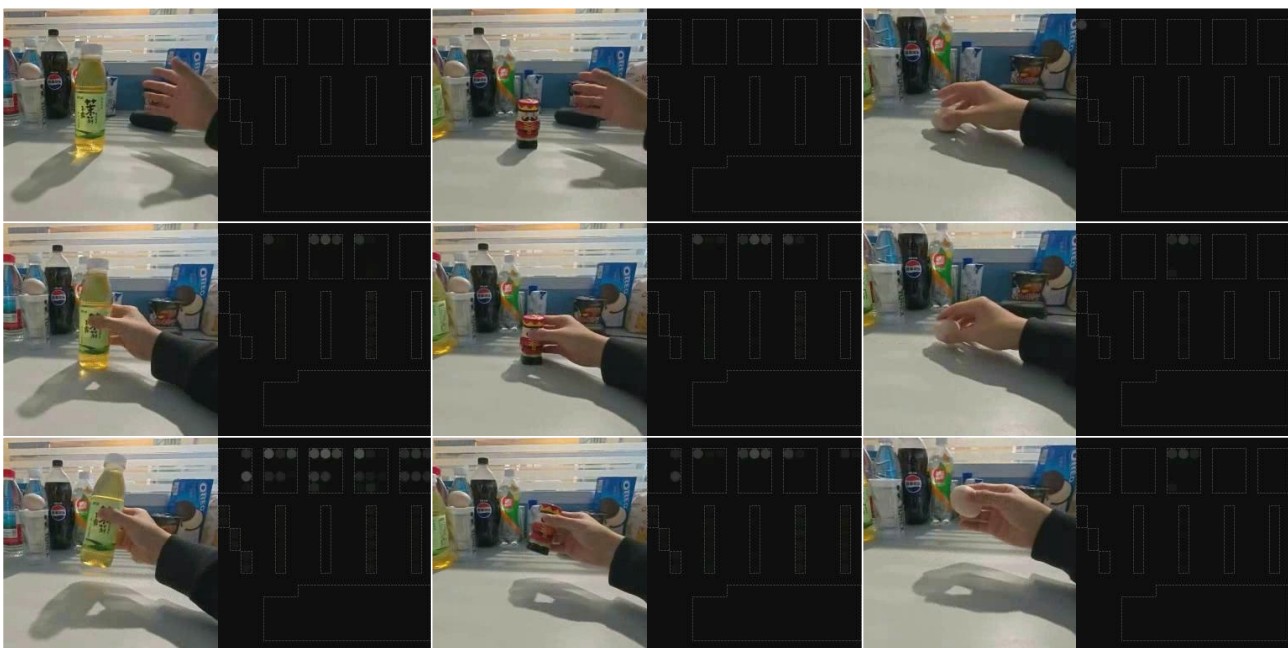

*Figure 14.* Qualitative robustness in the wild (3/3). We further evaluate EgoPressureDiff on object instances entirely absent from the training set, including an egg, JasmineGreenTea, and a NutcrackerFigurine. Even when encountering novel shapes and materials under unconstrained conditions, the model successfully infers reasonable contact geometries and pressure distributions, demonstrating strong object-level generalization.

| Ground Truth | Input Frame | EgoPressureDiff | Ground Truth | Input Frame | EgoPressureDiff |
|---|---|---|---|---|---|

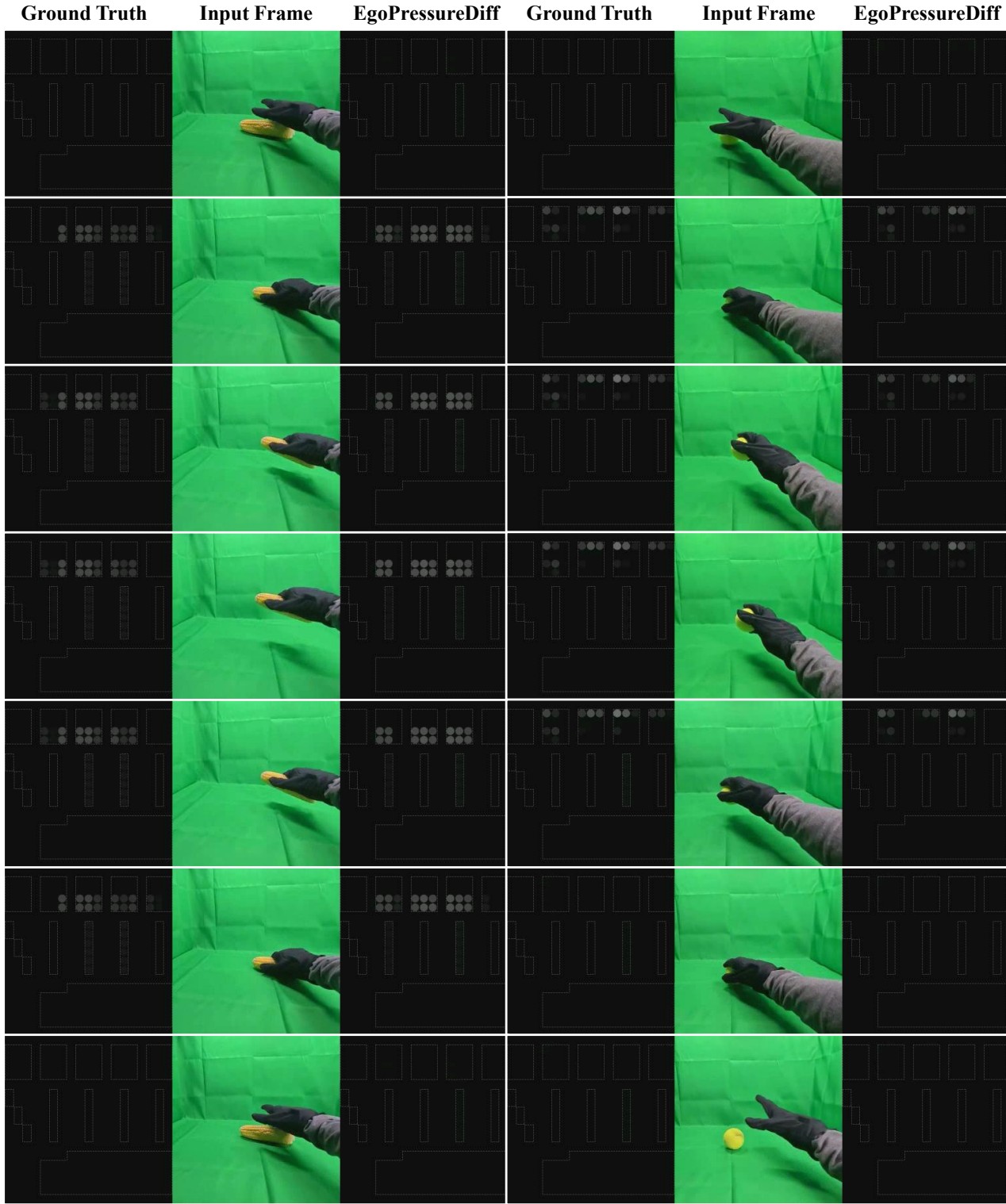

*Figure 15.* Additional Qualitative Results (Gloved-hand, Neck-mount). We visualize the continuous pressure prediction sequences on unseen objects under the Object-Held-Out protocol. **Left:** Grasping a Corn. **Right:** Grasping a Tennis Ball. The model generates temporally coherent heatmaps that accurately reflect the contact geometry of the curved surfaces.

| Ground Truth | Input Frame | EgoPressureDiff | Ground Truth | Input Frame | EgoPressureDiff |
| --- | --- | --- | --- | --- | --- |

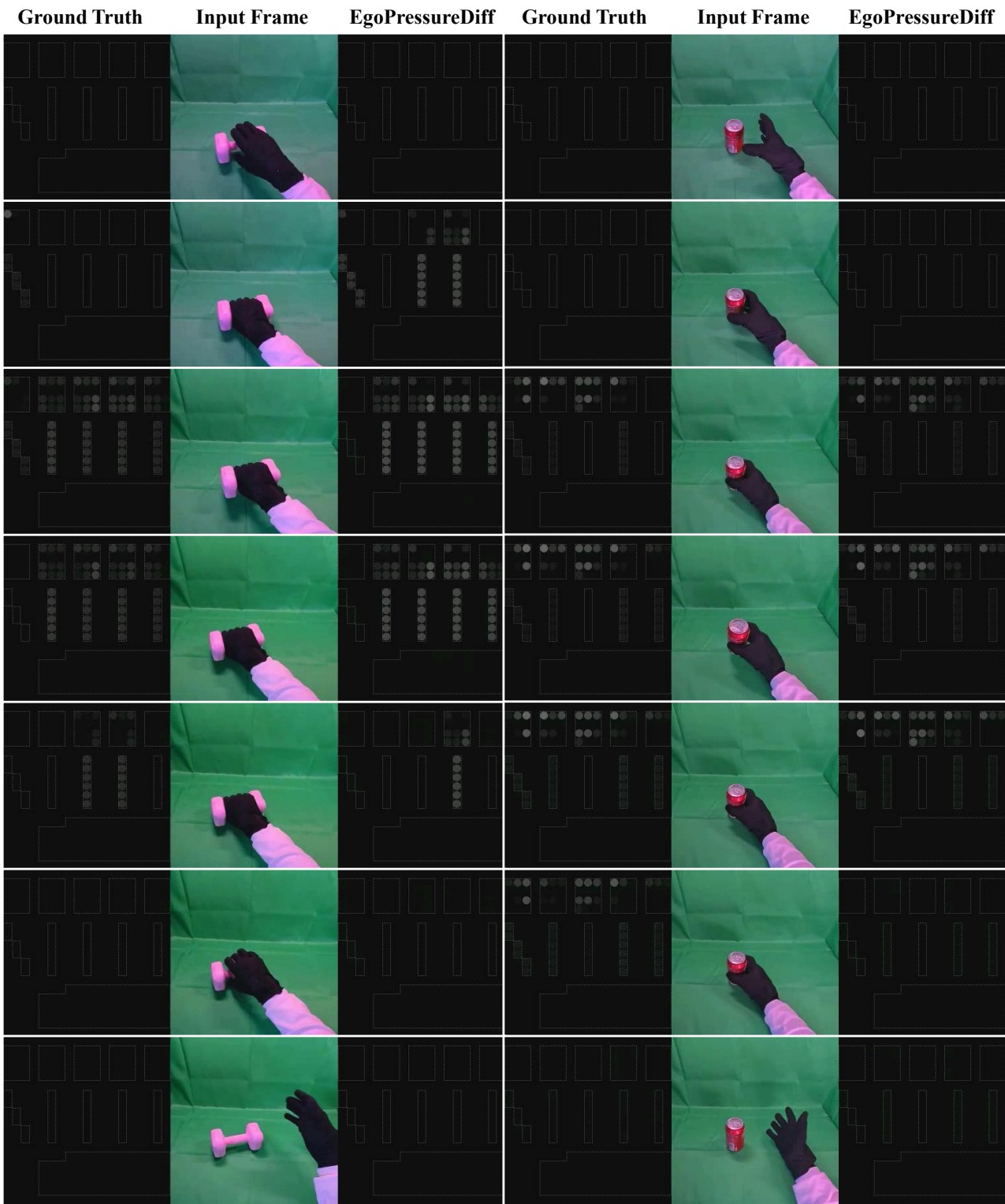

*Figure 16.* Additional Qualitative Results (Gloved-hand, Head-mount). Visualization of predictions on unseen objects from a head-mounted camera view. **Left:** Grasping a Dumbbell, showing high pressure on the palm and fingers corresponding to the heavy load. **Right:** Grasping a CocaCola-330ml. The model demonstrates robustness to viewpoint changes.

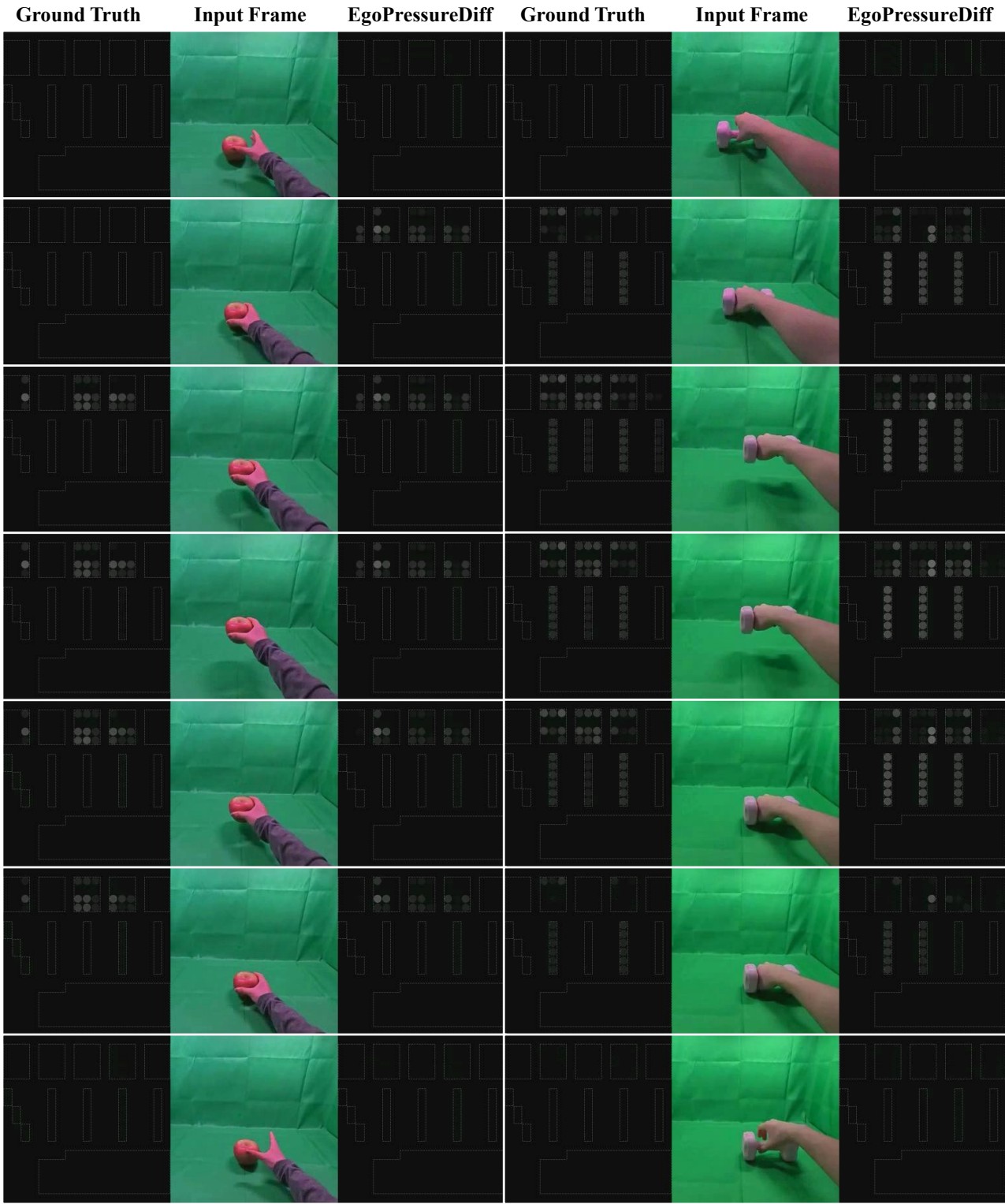

*Figure 17.* Additional Qualitative Results (Bare-hand). Visualization of the model's generalization to the bare-hand domain on unseen objects (Neck-mount). **Left:** Grasping an Apple. **Right:** Grasping a Dumbbell. Despite the appearance gap (no glove), the model accurately infers contact pressure.

