# OpenReview forum: "EgoTactile: Learning Grasp Pressure for Everyday Objects from Egocentric Video"
_ICML.cc/2026/Conference — ICML 2026 spotlight_

### Official Review · Reviewer_7rEY · 2026-03-09

**Soundness:** 4
**Presentation:** 4
**Significance:** 4
**Originality:** 3
**Overall Recommendation:** 5
**Confidence:** 3

**Summary:**

The paper aims to address the core challenges of data gap, occlusion-induced uncertainty and visual-physical ambiguities in vision-based full-hand grasp pressure estimation from egocentric video. It first constructs the EgoTactile benchmark dataset with a bare-hand transfer subset, and proposes the EgoPressureDiff, a conditional diffusion model paired with a PIFR layer for accurate pressure prediction.

**Compliance With Llm Reviewing Policy:**

Affirmed.

**Final Justification:**

All my questions have been resolved, so I will maintain my score.

**Key Questions For Authors:**

1. Could you supplement experimental results using flow matching or consistency models as the backbone, and report the FPS and corresponding performance?

**Limitations:**

Yes

**Strengths And Weaknesses:**

**Strengths:**
1. The paper proposes a new benchmark called EgoTactile, which provides a standardized platform for egocentric vision-based full-hand pressure estimation.
2. The proposed EgoPressureDiff effectively resolves the ill-posed problem of pressure estimation under occlusion and the visual-physical ambiguities in the task.
3. The experiments are sufficient, with complete ablation studies, verification, and clear, comprehensive analysis of both model performance and failure cases.
4. Overall, the paper is well-written, with a clear narrative and sufficient experimental validation.

**Weaknesses:**
1. EgoPressureDiff only achieves an inference speed of around 2.8 FPS, which may struggle to meet real-time requirements of practical deployment scenarios.

---

> ### Author Rebuttal · Authors · 2026-03-30
>
> We sincerely thank the reviewer for the constructive suggestion. We fully agree that inference speed is critical for practical deployment in VR, robotics, and interactive systems.
>
> # 1. W1 & Q1: Inference Speed and Alternative Backbones
>
> ## 1.1 Input resolution analysis
>
> We acknowledge that the current EgoPressureDiff is still slow. The main reason is that our method is built on a large SVD video diffusion backbone with iterative denoising.
>
> To control computation, we already use a relatively low resolution setting of 256×256 in both training and inference. Here, the resolution refers not only to the generated pressure heatmap, but also to the input RGB images and the segmentation maps.
> We choose this setting for two reasons.
> First, the target pressure field is a low dimensional physical signal that is finally mapped to 162 taxels, so it does not benefit from very fine visual detail. Second, the main challenge of this task comes from occlusion, invisible contact regions, and visual physical ambiguity, rather than low input resolution. Therefore, increasing resolution does not lead to proportional gains in useful physical evidence.
>
> To quantify this tradeoff, we additionally trained the model with different resolutions on the Object-Held-Out split.
>
> *Table 1. Performance and speed under different RGB, mask, and pressure map resolutions.*
>
> |Resolution|C-IoU(%)↑|V-IoU(%)↑ |MAE(N)↓|CoP↓ |FPS|
> |-|:-:|:-:|:-:|:-:|:-:|
> |128×128|51.8|35.4|3.9|3.6|5.0|
> |256×256|56.3|38.9|3.4|3.1|2.8|
> |512×512|56.9|39.4|3.3|3.0|0.9|
>
> As shown in Table 1, 128×128 is faster but causes clear degradation in contact localization and pressure magnitude estimation. In contrast, 512×512 brings only marginal gains but significantly increases cost. This further suggests that the dominant error source is not only missing visual detail, but also occlusion and ambiguous visual to physical mapping. Therefore, 256×256 is a balance choice between accuracy and efficiency.
>
> ## 1.2 Why we use LCM instead of Flow Matching
>
> To address the reviewer’s concern, we performed additional acceleration experiments with consistency models instead of flow matching.
>
> The main reason is compatibility with the current model. EgoPressureDiff is an SVD based conditional diffusion model, and its mask branch, prototype conditioning, and PIFR are coupled with the current denoising pipeline. LCM can directly distill the original 25 step teacher into a few step student while preserving the same backbone and conditioning design.
>
> By contrast, flow matching would require replacing the current diffusion objective with velocity field regression and redesigning how the conditioning modules interact with the generation process. To complete the additional experiments within the rebuttal period, we therefore choose the LCM, which can be integrated much more efficiently.
>
> ## 1.3 LCM distillation and results
>
> For LCM, we keep the same training setup as the original model, including 256×256 resolution, 16 frame clips, and the same data split. We use the original 25 step EgoPressureDiff as the teacher and train few step students on the same training set.
>
> During distillation, we freeze the VAE and CLIP encoders, and distill the 3D U-Net, the Mask Encoder, and the PIFR related conditioning modules. We keep the same conditioning design and task definition, and only reduce the sampling steps through distillation.
> We evaluate LCM students with different sampling steps on the object held out split.
>
> *Table 2. Accuracy and speed tradeoff after LCM distillation.*
>
> |Method|Steps|C-IoU(%)↑|V-IoU(%)↑|MAE(N)↓|CoP↓|FPS|
> |-|:-:|:-:|:-:|:-:|:-:|:-:|
> |EgoPressureDiff|25|56.3|38.9|3.4|3.1|2.8|
> |LCM student|8|54.9|37.3|3.7|3.3|6.9|
> |LCM student|4|52.1|35.2|4.4|3.8|10.3|
>
> LCM provides a clear inference speed gain. The 8 step student improves speed from 2.8 FPS to 6.9 FPS, which is about 2.5x faster, with only a small performance drop. The 4 step student further reaches 10.3 FPS, about 3.7x faster. The main error increase in the few step setting appears in pressure magnitude calibration and contact localization.
>
> ## 1.4 Future work
>
> We thank the reviewer again for this important suggestion. The results show that LCM is a practical method for the current EgoPressureDiff framework, improving inference speed to 6.9 FPS and 10.3 FPS while preserving competitive performance. We will add these results in the revised paper.
>
> And, we agree that flow matching is a promising direction. We currently adopt an SVD backbone because, on a real world dataset of this scale, SVD style U-Net diffusion is easier to optimize. In comparison, DiT based video diffusion models usually require larger scale data to fully demonstrate their stronger capacity. In future work, we plan to expand the dataset through simulation combined with real capture, and then migrate to a DiT backbone with flow matching training, with the goal of further improving downstream usability.

---

> > ### Author Rebuttal · Reviewer_7rEY · 2026-04-02
> >
> > Thank the author for the detailed replies. All my questions have been resolved, and **I will keep my score as Accept.**

---

### Official Review · Reviewer_TT9C · 2026-03-10

**Soundness:** 2
**Presentation:** 2
**Significance:** 3
**Originality:** 3
**Overall Recommendation:** 4
**Confidence:** 3

**Summary:**

This paper introduces EgoTactile, a benchmark for predicting full-hand grasp pressure on everyday 3D objects from egocentric RGB video. It collects paired egocentric clips and glove pressure signals for 63 objects, and additionally provides a bare-hand transfer subset via weakly paired supervision to encourage generalization beyond instrumented hands. Besides, it establishes a discriminative baseline and proposes a conditional diffusion approach. Experiments on object-held-out and subject-held-out protocols show consistent improvements over prior vision-based pressure estimation baselines.

**Compliance With Llm Reviewing Policy:**

Affirmed.

**Final Justification:**

The authors have addressed my concerns, so I will maintain my positive score.

**Key Questions For Authors:**

1. Can the authors provide additional quantitative or qualitative analyses that validate the consistency of the bare-hand data collection protocol (i.e., how well the off-camera gloved pressure corresponds to the on-camera bare-hand interaction)? For instance, do you measure temporal alignment error, contact area agreement, or repeatability across trials?

2. Can the authors show more results under varied hand poses and interaction types (e.g., two-finger pinch, palm support/carry, or other non-grasp manipulations)? Without retraining, how well does the model generalize to such unseen poses/actions, and where does it fail?

**Limitations:**

yes

**Strengths And Weaknesses:**

## Strengths

1. The paper is logically structured and easy to follow. It effectively highlights the significance and challenges of egocentric 3D grasp prediction, establishing a strong motivation for the proposed work.

2. EgoTactile provides a valuable benchmark direction by targeting a gap between vision-only interaction understanding and dense tactile supervision, focusing on 3D objects and full-hand contact under egocentric occlusion.

## Weaknesses

1. In the bare-hand protocol, the visible interaction is performed with a bare hand, while the pressure labels are collected from an off-camera gloved hand. Even with metronome guidance, the recorded pressure may not accurately correspond to the visible hand’s true forces and contact patterns. This mismatch could be exacerbated when the same subject operates both hands simultaneously, which may lead to inconsistent timing and grasp execution.

2. The in-the-wild experiments lack ground truth and therefore only provide qualitative results. The paper could strengthen this section by reporting quantitative results in more realistic environments where ground truth is still available. For example, gloved-hand captures with pressure labels but with varied, cluttered backgrounds and lighting.

3. Across objects and subjects, the experiments seem to use a largely fixed grasp configuration. Since a key contribution is full-hand pressure estimation, the current evaluation does not clearly demonstrate the advantages enabled by full-hand coverage under diverse grasp poses and interaction types.

---

> ### Author Rebuttal · Authors · 2026-03-30
>
> We thank the reviewer for the careful comments and helpful suggestions.
>
> # 1. W1&Q1: Consistency of the Bare Hand Protocol
>
> We agree that the bare hand protocol is not aligned at the same level as the gloved hand protocol. Under current hardware limits, it is still a practical way to obtain natural bare hand video with pressure labels at scale. Alternatives such as sEMG give only coarse force cues$^{[1]}$, while object mounted sensors are hard to scale$^{[2]}$.
>
> To test consistency, we ran a new dual glove study that simulates the bare hand protocol. Both hands wear pressure gloves. One hand is visible in the camera, and the other grasps the same object outside the view. We use 3 subjects × 5 held out objects × 10 repeats = 150 clips. We compare pressure labels from the two hands, not model prediction versus GT. The results show onset and release gaps of about 105 to 118 ms (less than 2 frames at 15 FPS), contact area agreement above 82%, and small CoP and force gaps. This indicates the weak pairing is imperfect, but not random.
>
> *Table 1. Dual-glove consistency under bare-hand protocol.*
>
> |Object|ΔContact Onset(ms)↓|ΔRelease(ms)↓|ΔContact Duration(ms)↓|C-IoU(%)↑|CoP↓|MAE(N)↓|
> |:-|:-:|:-:|:-:|:-:|:-:|:-:|
> |Apple|101|116|149|82.6|2.4|5.1|
> |CocaCola330ml|94|108|136|84.1|2.2|4.7|
> |Corn|126|139|181|79.8|2.8|6.3|
> |Dumbbell|87|96|129|86.3|2.0|4.2|
> |TennisBall|118|131|174|80.5|2.7|5.8|
> |**Overall**|105±31|118±36|154±42|82.7±5.6|2.4±0.7|5.2±1.6|
>
>
> We also measured cross trial repeatability on 1 subject, 3 objects, and 30 trials per object. For each trial, we compare its mean contact pressure map against the average of the other 29 trials of the same object.
>
> *Table 2. Cross-trial repeatability under bare-hand protocol.*
>
> |Object|ΔContact Duration(ms)↓|C-IoU(%)↑|MAE(N)↓|
> |:-|:-:|:-:|:-:|
> |Apple|148±39|86.4±4.2|4.6±1.0|
> |CocaCola330ml|131±34|88.7±3.8|4.0±0.9|
> |Dumbbell|156±43|85.1±4.6|4.9±1.1|
> |**Overall**|145±39|86.7±4.2|4.5±1.0|
>
> These results support that the bare hand subset is weakly paired but consistent enough to serve as a transfer benchmark. We will add these analyses and clarify the limitation more clearly.
>
> *[1] Seo, Kyung Jin, et al. "Posture-Informed Muscular Force Learning for Robust Hand Pressure Estimation." NeurIPS. 2024.*
>
> *[2] Pham, Tu-Hoa, et al. "Hand-object contact force estimation from markerless visual tracking." IEEE TPAMI 40.12 (2017): 2883-2896.*
>
> # 2. W2: Quantitative Evaluation in Realistic Scenes
>
> We agree that the in the wild section should include quantitative results. We collected 30 gloved hand clips across 10 everyday scenes with cluttered backgrounds and unconstrained lighting. All objects are unseen during training. PressureVision and EgoPressureFormer drop a lot, while EgoPressureDiff shows a much smaller decrease. One plausible explanation is that the strong prior from the large SVD backbone helps the model infer physically plausible contact patterns even when background and illumination vary.
>
> *Table 3. Quantitative results in realistic scenes.*
>
> |Method|Setting|C-IoU(%)↑|V-IoU(%)↑|MAE(N)↓|CoP↓|
> |-|-|-|-|-|-|
> |PressureVision|Controlled |24.5|16.8|9.2|12.5|
> ||In-The-Wild|13.4|8.3| 14.0|16.0|
> ||Δ(Change)|-11.1|-8.5|+4.8|+3.5|
> |EgoPressureFormer|Controlled|36.8|26.5|6.2|7.5|
> ||In-The-Wild|23.7|16.1|8.9|10.7|
> ||Δ(Change)|-13.1|-10.4|+2.7|+3.2|
> |EgoPressureDiff|Controlled|56.3|38.9|3.4|3.1|
> ||In-The-Wild|49.6|33.7|4.6|5.4|
> ||Δ(Change)|-6.7|-5.2|+1.2|+2.3|
>
> # 3. W3&Q2: Generalization to Unseen Poses and Interaction Patterns
>
> We appreciate the reviewer’s thoughtful suggestion. In the original collection, we did not force a fixed grasp, but natural behavior still leads to similar patterns. We therefore built a new unseen pose and interaction test set with 40 clips under the gloved hand protocol on the 3 held out objects. The interactions include, but are not limited to, palm press and roll, palm support, hook lift, toss and catch, and palm base support.
>
> *Table 4. Generalization to unseen patterns.*
>
> |Method|Setting|C-IoU(%)↑|V-IoU(%)↑|MAE(N)↓|CoP↓|
> |-|-|-|-|-|-|
> |PressureVision|Natural Patterns|24.5|16.8|9.2|12.5|
> ||Unseen Patterns|9.8|5.9|15.8|17.4|
> ||Δ(Change)|-14.7|-10.9|+6.6|+4.9|
> |EgoPressureFormer|Natural Patterns|36.8|26.5|6.2|7.5|
> ||Unseen Patterns|16.7|10.8|10.9|13.2|
> ||Δ(Change)|-20.1|-15.7|+4.7|+5.7|
> |EgoPressureDiff|Natural Patterns|56.3|38.9|3.4|3.1|
> ||Unseen Patterns|31.8|21.4|7.1|7.8|
> ||Δ(Change)|-24.5|-17.5|+3.7|+4.7|
>
> All methods drop clearly in this setting. This is likely because the current inputs do not explicitly encode detailed hand pose or object 6D pose, so the models have limited awareness of rare interaction geometry.
> While EgoPressureDiff can still produce coarse pressure predictions for grasping actions during the interaction process, it struggles with entirely unfamiliar interaction modes. We will add these results to the revised paper. Detailed visual examples can be found in *Section G* of the [project page](https://egotactile.github.io/).

---

> > ### Author Rebuttal · Reviewer_TT9C · 2026-04-01
> >
> > I would like to thank the authors for their detailed rebuttal. My previous questions and concerns have been addressed.

---

### Official Review · Reviewer_ZZd8 · 2026-03-11

**Soundness:** 3
**Presentation:** 3
**Significance:** 4
**Originality:** 3
**Overall Recommendation:** 5
**Confidence:** 2

**Summary:**

The paper contributes to how hand grasp pressure map can be generated from the ecocentric video. The contributions of the paper are as follows: EgoTactile, a benchmark dataset for pressure that was collected by the authors. The two baseline approaches to pressure  map generation are created: EgoPressureDiff - generative model and EgoPressureFormer a discriminative approach. EgoPressireDiff appears to have state-of-the-art results.

**Compliance With Llm Reviewing Policy:**

Affirmed.

**Final Justification:**

my questions and comments were fully resolved. I am in favor in accepting this work, as before.

**Key Questions For Authors:**

Was the audio recorded as well in the dataset?

The paper specifically targets a human hand? Could this approach transfer to sensors in a robot’s hand ?

Why not add a more recent baseline ( PressureVision++) from 2024. The original PressureVision baseline could be outdated as it is from 2022.

Did adding demographic information to the prompts make any difference (e,g, improve the accuracy vs. no demographics baseline). It is difficult to understand how the age and weight of the subject affects the grasp.

What is the difference between the collected data and ActionSense dataset ? Action sense seems to have even better hand resolution.

**Limitations:**

The paper would be benefit from limitations sections in the main body of the paper. Some limitations are discussed in the appendix as failed cases.

**Strengths And Weaknesses:**

***Soundness***

The validation and implementation are technically sound. The system and dataset are well thought out.
The best validation for the work could be using the simulation data generated with EgoTactile and demonstrating an improvement on real robot manipulation tasks. I recognize this would be out of scope for this paper, but could be a good next step.

Some comments are below:

There might be differences in the pressure if the subject is grasping with their left or right hand, depending on which is their dominant hand (e.g., right-handed). It would be good to provide information regarding which hand was bare and gloved.

What glove sensors were used ? E.g., who manufactured the glove? It is important information for reproducibility.

An idea would be to add microphones and use the sound as well, as tactile interactions often make sounds depending on the object. I wonder if the sounds could make the heatmap more accurate

It could be useful to include random guessing results in Table 4.

***Presentation***

The paper is understandable and the work seems to be reproducible. The appendix is well used to include important information.
In figure 3, I would label the U-Net as SVD, as it is not clear what part of the pipeline is SVD. Also, I assume that U-Net is being finetuned, however the fire image across the whole thing makes it looks like it's being trained from scratch. I would add a label to the diagram.

***Significance***

Paper is of high significance.
The paper tackles an important topic of how to generate realistic data such as pressure maps from grasps. This would allow improving fidelity of robotics simulations and world models. The paper contribution in this area is twofold: First, a novel validation dataset with Ego centric view point and tactile glove grasps, collected in real-world. Collecting data is an extremely valuable contribution, as it provides ground truth.  Second, a generative model, that appears to be state of the art (compared to PressureVision model)

***Originality***

The originality is good.
The overall concept of generating pressure maps from vision has been already explored such as the PressureVision paper in 2022 as well as few others. The originality of the paper is employing the diffusion generative approach with physical priors conditioning. The insights of this is that the diffusion approach works well for this task, as it is better in dealing with ambiguities. Also, the paper introduces a new dataset collected with humans. Overall, the dataset seems to be similar to ActionSense, as seen in Table 8.  Benchmark dataset is certainly important for the future research in this area.

---

> ### Author Rebuttal · Authors · 2026-03-30
>
> We deeply appreciate the constructive feedback of the reviewer, which will undoubtedly strengthen the paper.
>
> # 1. C1: Hand Dominance & Bare/Gloved Setup
> We completely agree that hand dominance affects grasp pressure. To standardize this, all 12 participants in our dataset are right-handed and always used their dominant hand for pressure generation.
> * **Gloved Setting:** Participants used their gloved right hand.
> * **Bare-hand Setting:** Participants grasped the object with their bare left hand (in-view) while synchronously grasping an identical object with their gloved right hand (out-of-view).
> We will clarify more details in the dataset section.
>
> # 2. C2: Glove Manufacturer
> The glove is from JQ-Industries Technology. Details will be added to the Appendix.
>
> # 3. C3 & Q1: Audio Modality
> Audio was not recorded in the current dataset, but this suggestion is brilliant. Acoustics provide strong physical cues: the sound frequency distinguishes an empty vs. a full plastic bottle, and audio peaks precisely indicate contact onset/offset. Integrating a microphone to capture interaction acoustics could help resolve visual-physical ambiguities. We will highlight audio-tactile integration as a promising direction for future work.
>
> # 4. C4: Random Guessing Baseline
> We implemented a marginal-probability sampler as a Random Guesser. Table 1 confirms our models learn physical correlations, not just dataset statistics. We will add these results and the details of this Random Guessing Baseline in the revised paper.
>
> *Table 1. Comparison with Random Baseline.*
>
> | Method|Temp Acc.↑|C-IoU↑|V-IoU↑|MAE↓|CoP↓|
> |:-|:-|:-|:-|:-|:-|
> | Random Guesser |41.8%|7.6%|4.1%|15.4|30.9|
> | PressureVision |65.2%|24.5%|16.8%|9.2|12.5|
> | EgoPressureFormer |84.5%|36.8%|26.5%|6.2|7.5|
> | EgoPressureDiff |96.4%|56.3%|38.9%|3.4|3.1|
>
> # 5. Q2: Transfer to Robot Hands
> This approach holds strong potential for robotics, as the tactile glove we utilized is also designed for humanoid dexterous hands. A bottleneck in Visual-Tactile-Language-Action policies is the lack of paired data. Applying EgoPressureDiff to uninstrumented egocentric datasets generates dense tactile pseudo-labels from video, enabling robots to learn human-like force strategies via large-scale imitation learning.
>
> # 6. Q3: PressureVision++ Baseline & Temporal Modeling
> We thank the reviewer for highlighting this recent and relevant baseline. We did not include PressureVision++ for two reasons.
> First, PV++ relies on weak supervision, which is incompatible with our fully supervised dataset.
> Second, Removing its weak-supervision components reduces it to a PV-like architecture, which we already compare against.
> Additionally, we emphasize a fundamental difference in task formulation: prior works (PV, PV++, EgoPressure) are single-frame spatial models. In contrast, our proposed baselines utilize video backbones to explicitly model the temporal dynamics of grasping. This temporal continuity is crucial for pressure estimation.
>
> # 7. Q4: Impact of Demographic Information
> To answer this question, we additionally include the corresponding ablation.
>
> As shown in Table 2, object info remains the dominant factor, while subject info acts more as a complementary individual prior. In Object-Held-Out, removing subject info causes only a small drop, indicating limited influence when object info is available. In Subject-Held-Out, the drop is larger, suggesting subject info helps generalize to unseen participants.
>
> We believe the current gain from subject demographics is still limited because the dataset contains only 12 participants, with a modest range of demographic variation. With more participants and larger inter-subject diversity, we expect the benefit of demographic information to become more evident.
>
> *Table 2. Effect of Demographic Information in Text Conditioning.*
>
> | Variant|Obj-H-O V-IoU↑|Obj-H-O MAE↓|Subj-H-O V-IoU↑|Subj-H-O MAE↓|
> |-|-:|-:|-:|-:|
> | Full text conditioning |38.9|3.4|34.0|5.8|
> | *w/o* subject info |38.2|3.8|30.9|6.6|
> | *w/o* object info |29.9|4.9|28.4|6.8|
> | *w/o* text |30.3|5.1|28.0|7.1|
>
>
> # 8. Q5: Differences from ActionSense
> We highly respect ActionSense and will explicitly detail our differences in the revised paper:
> * Crucial Sensor Coverage: ActionSense uses "open finger" gloves (no sensors on the fingertips). Because fingertips bear the primary force during grasping, EgoTactile's continuous, full-hand coverage is critical for physical grasp pressure estimation.
> * Bare-Hand Transfer: ActionSense only contains gloved interactions. We provide a unique weakly-paired bare-hand subset to enable real-world, sensor-free transfer.
> * Environment: ActionSense captures noisy kitchen data, whereas EgoTactile uses a controlled green-screen setting to isolate visual-pressure correlations.
>
> # 9. Presentation & Limitations
> We thank the reviewer for the helpful suggestions on presentation. We will incorporate all presentation suggestions and add a dedicated limitations section.

---

> > ### Author Rebuttal · Reviewer_ZZd8 · 2026-04-03
> >
> > Thank you for addressing my questions in detail. I don't have any follow up questions.

---

### Official Review · Reviewer_admF · 2026-03-13

**Soundness:** 3
**Presentation:** 3
**Significance:** 3
**Originality:** 3
**Overall Recommendation:** 4
**Confidence:** 3

**Summary:**

This paper proposes EgoTactile, a new benchmark for estimating full-hand grasp pressure from egocentric video. The dataset consists of egocentric videos with hand pressure annotations, primarily captured with gloved hands for accurate pressure measurement, along with a bare-hand subset to enable generalization. Built on this benchmark, the paper also introduces two methods for egocentric hand pressure estimation: (1) EgoPressureFormer, a discriminative baseline, and (2) EgoPressureDiff, a generative baseline based on a video diffusion backbone. Experiments show that these methods outperform existing baselines and demonstrate potential for in-the-wild generalization.

**Compliance With Llm Reviewing Policy:**

Affirmed.

**Final Justification:**

The authors have resolved my questions. Therefore, I will maintain my positive rating.

**Key Questions For Authors:**

Will the collected dataset be published?

**Limitations:**

Yes.

**Strengths And Weaknesses:**

**[Strengths]**

1. **Well-designed benchmark and task.**

The dataset collection is carefully conducted (e.g., including both gloved-hand and bare-hand settings as well as diverse participants), and the models built on this dataset are well designed. Especially, the generative formulation is reasonable for handling uncertainty in this problem.

2. **Good writing quality.**

The paper clearly explains the proposed ideas and articulates the advantages over existing benchmarks and methods.

3. **Strong experimental results.**

The proposed methods outperform existing baselines. The paper also includes ablation studies analyzing the contributions of individual components and demonstrates in-the-wild generalization results (though I have one question about the generalization ability; see Weakness 1 below).

**[Weaknesses]**

1. **Limited background diversity in the dataset.**

The collected dataset appears to contain only a lab-style background. Could this negatively affect the downstream model’s generalizability?

2. **Reproducibility concerns.**

The work heavily relies on a newly collected dataset. Whether the dataset will be publicly released is important for ensuring reproducibility.

---

> ### Author Rebuttal · Authors · 2026-03-30
>
> We sincerely thank the reviewer for the careful reading and the encouraging assessment of our paper. The reviewer's questions on background diversity and reproducibility are important, and we will clarify them below.
>
> # 1. W1: Background diversity / generalization
>
> ## 1.1 The controlled setup ensures reliable supervision
>
> We agree that the supervised benchmark data are collected in a controlled lab environment. This is a deliberate design choice. Since this work introduces a new task, namely estimating dense full hand grasp pressure from egocentric video, our first goal is to establish whether pressure can be inferred from visual input under accurate tactile supervision. A controlled setup helps reduce irrelevant factors and makes the supervision trustworthy.
>
> ## 1.2 The benchmark already reduces overfitting
>
> Although the labels are collected in a controlled environment, we already include several mechanisms to reduce overfitting. During data collection, we randomize lighting and object poses, and we use both head-mounted and neck-mounted camera views (*Sec. 4.1; Fig. 2c-d*). During evaluation, we include Object-Held-Out and Subject-Held-Out protocols, which test transfer to unseen objects and unseen participants (*Sec. 4.2; Table 3; Table 4*). We also include a bare-hand subset to test appearance shift from instrumented gloved hands to natural hands (*Sec. 4.1, Bare-hand set; Sec. 6.3; Table 5; Fig. 5, bottom row*). In addition, the paper already presents qualitative in-the-wild examples under cluttered backgrounds and changing lighting (Sec. 6.4; Appendix E.1.2; Figs. 12-14). We will make these points more explicit in the revision.
>
> ## 1.3 Additional quantitative in the wild experiment
>
> To directly address the reviewer’s concern, we additionally construct a quantitative in the wild test set. Specifically, we record 30 egocentric gloved hand videos across 10 everyday scenes, including the kitchen, bathroom sink area, office desk, dining room, living room, and bedroom, with cluttered backgrounds and unconstrained lighting. The grasped objects are all unseen during training and cover common household items such as a mug, water bottle, bowl, shampoo bottle, remote control, and stapler.
>
> *Table 1.  Quantitative results in realistic gloved-hand scenes (In-The-Wild) under Object Held Out protocol.*
>
> |Method|Setting|C-IoU(%)↑|V-IoU(%)↑|MAE(N)↓|CoP↓|
> |-|-|-|-|-|-|
> |**PressureVision**|Controlled|24.5|16.8|9.2|12.5|
> ||In-The-Wild|13.4|8.3|14.0|16.0|
> ||*Δ (Change)*|*-11.1*|*-8.5*|*+4.8*|*+3.5*|
> |**EgoPressureFormer**|Controlled|36.8|26.5|6.2|7.5|
> ||In-The-Wild|23.7|16.1|8.9|10.7|
> ||*Δ (Change)*|*-13.1*|*-10.4*|*+2.7*|*+3.2*|
> |**EgoPressureDiff**|Controlled|**56.3**|**38.9**|**3.4**|**3.1**|
> ||In-The-Wild|**49.6**|**33.7**|**4.6**|**5.4**|
> ||*Δ (Change)*|***-6.7***|***-5.2***|***+1.2***|***+2.3***|
>
>
>
> The results show a clear trend. PressureVision and EgoPressureFormer suffer large drops compared with the gloved hand Object-Held-Out benchmark (Controlled), while EgoPressureDiff shows only a mild decrease. These results suggest that EgoPressureDiff generalizes better under real scene changes. One plausible explanation is that the strong prior from the large SVD backbone helps the model infer physically plausible contact patterns even when background and illumination vary. This is also consistent with the conclusion suggested by Table 5 that EgoPressureDiff exhibits stronger transfer ability than discriminative baselines.
>
> ## 1.4 The current evidence supports robustness
>
> We agree that broader real world coverage remains an important limitation and future direction. In the revision, we will clearly distinguish controlled supervision from downstream deployment, and we will add the new quantitative in the wild experiment to make the generalization discussion more complete. We also believe that adding more unconstrained data in future work can further improve the robustness of EgoPressureDiff.
>
> # 2. W2 and Q1: Dataset release / reproducibility
>
> Yes. The dataset and code are anonymously released during review.
>
> Project page: [https://egotactile.github.io/](https://egotactile.github.io/)
>
> Dataset: [https://huggingface.co/datasets/icml-2026-submission/EgoTactile](https://huggingface.co/datasets/icml-2026-submission/EgoTactile)
>
> Code: [https://anonymous.4open.science/r/EgoPressureDiff-4DCB/README.md](https://anonymous.4open.science/r/EgoPressureDiff-4DCB/README.md)
>
> We fully agree with the reviewer that public release is important for this task, especially because the work is built on a newly collected dataset. This release status will be stated more explicitly in the revision.

---

> > ### Author Rebuttal · Reviewer_admF · 2026-04-04
> >
> > I appreciate the authors’ response, including the additional in-the-wild experimental results. My initial questions have been resolved.

---

### Decision · Program_Chairs · 2026-04-30

**Decision:**

Accept (spotlight)

**Comment:**

All four reviewers recommend clear acceptance. The author rebuttals have successfully addressed the reviewer concerns. The proposed topic is important to the field, the work is worth sharing with the ICML community. ACs welcome the work.